# In situ measurement of $CO_2$ and $CH_4$ from aircraft over northeast China and comparison with OCO-2 data

Xiaoyu Sun[1,2], Minzheng Duan[*1,2,3], Yang Gao[4], Rui Han[2,5], Denghui Ji[1,2], Wenxing Zhang[1], Nong Chen[6], Xiangao Xia[1,2], Hailei Liu[3], Yanfeng Huo[7]

[1]LAGEO, Institute of Atmospheric Physics, Chinese Academy of Sciences, 100029 Beijing, China
[2]College of Earth and Planetary Sciences, University of Chinese Academy of Sciences, Beijing,100049, China
[3]Chengdu University of Information Technology, Chengdu, 610225, China,
[4]China Meteorological Administration, Beijing, 100089, China
[5]ICCES, Institute of Atmospheric Physics, Chinese Academy of Sciences, Beijing, 100029, China
[6]Heilongjiang Meteorological Bureau, Harbin, 150001, China
[7]Anhui Meteorology Service, Hefei, 230061, China

*Corresponding to:* Minzheng Duan (dmz@mail.iap.ac.cn)

**Abstract.** Several satellites have been launched to monitor the increasing concentrations of greenhouse gases, especially $CO_2$ and $CH_4$ in the atmosphere, through back-scattered hyperspectral radiance in the shortwave infrared (SWIR) band. The vertical profiles of greenhouse gases and aerosol could strongly affect the results from these instruments. To investigate the effects of the vertical distribution of $CO_2$ on the uncertainty of SWIR satellite retrieval results, we conducted observations of the vertical profiles of $CO_2$, $CH_4$, and aerosol particles at 0.6–7 km above sea level using a Beechcraft King Air 350ER in Jiansanjiang (46.77°N, 131.99°E), Heilongjiang province, Northeast China, on 7-12 August, 2018. The profiles from this aircraft captured a decrease in $CO_2$ from 2 km to the minimum altitude due to the absorption of vegetation at the surface in summer. $CH_4$ measurements showed about 0.2 ppm increase from 2.0 to 0.6 km in 10 August, which may result from emissions from the large area of paddy fields below, and a constant mole fraction between 1.951 and 1.976 ppm was recorded at 2 km and above in three days. Comparison of $CO_2$ profiles from a new version of the carbon cycle data assimilation system Tan-Tracker (v1), retrievals from OCO-2 and aircraft measurements was conducted. The results from OCO-2 and the assimilation model system Tan-Tracker captured the vertical structure of $CO_2$ above 3 km, whereas below 3 km, the values from OCO-2 and Tan-Tracker model were lower than those from in situ measurements. Column-averaged $CO_2$ volume mole fractions calculated from in situ measurements showed biases of $-4.68 \pm 0.44$ ppm ($-1.18\% \pm 0.11\%$) compared to OCO-2 retrievals.

## 1 Introduction

Global warming due to greenhouse gases (GHGs) has become one of the most urgent and widely studied issues faced by scientists in recent years. The Fifth Assessment Report of the Intergovernmental Panel on Climate Change (IPCC) noted that the global average temperature has increased by 0.85°C over the period of 1880–2012. GHGs, especially the increasing $CO_2$ levels in the atmosphere related to anthropogenic activities, are blamed for global warming, because they absorb and emit

radiant energy within the thermal infrared range. Emission of $CO_2$ from fossil fuel combustion and industrial processes has contributed about 78% of the total GHG emissions increase from 1970 to 2010 (IPCC, 2015). Accurate measurement of $CO_2$ concentrations and their spatial and temporal variations in the atmosphere is essential for estimation of sources and sinks in

regional and global models (Patra et al., 2005a, 2005b; Zhang et al., 2008). The Global Atmospheric Watch program (http://www.wmo.int/gaw) coordinates the systematic observation and analysis of GHGs and other trace substances, providing an important source of local and global GHG data. However, ground-based and in situ measurements near the surface can only provide information about the lower atmosphere, and are insufficient for analysis of total-column GHGs, which exhibits variations in both vertical and horizontal directions. Over the past few years, several satellites, including the Greenhouse Gases

Observing Satellite (GOSAT, launched in January 2009), Second Orbiting Carbon Observatory, (OCO-2, launched in 2014), and TanSat (launched in 2016), have been launched into space to monitor $CO_2$ by observing back-scattered hyperspectral radiance in shortwave infrared (SWIR) wavelength, which can provide global coverage of the column-averaged dry-air mole fraction of $CO_2$ ($X_{CO2}$). Studies have shown that, given a 1–2 ppm accuracy of $X_{CO2}$, the use of space-borne instrument data can reduce the uncertainties in regional ($8° \times 10°$ footprint) estimation of $CO_2$ sources and sinks (Rayner and O'Brien, 2001).

In addition, $CO_2$ vertical profiles in the 5–25 km altitude range can be obtained using limb viewing space-borne sounders such as the Atmospheric Chemistry Experiment Fourier Transform Spectrometer (ACE-FTS, launched in August 2003). Foucher et al. (2009) reported the feasibility and difficulties of obtaining vertical $CO_2$ profiles using this method.

To validate and calibrate the $X_{CO2}$ data from satellite measurement products, the Total Carbon Column Observing Network (TCCON), a network of ground-based solar Fourier transform spectrometers operating in the SWIR spectral region was

established (Wunch et al., 2011). Several studies have been conducted to determine $X_{CO2}$, the column-averaged $CH_4$ volume mole fraction ($X_{CH4}$), and the column-averaged volume mole fractions of other trace gases ($X_{gas}$) from TCCON data, which have shown good accuracy (Hedelius et al., 2017; Mendonca et al., 2019). In addition, commercial mobile solar-viewing near-infrared spectrometers of lower resolution than the TCCON instruments, such as Bruker$^{TM}$ EM27/SUN, show potential for measurement of $X_{gas}$ with an acceptable bias range (Hedelius et al., 2016).

Retrieval accuracy is affected by knowledge of the vertical distribution of aerosols and $CO_2$. Vertical profiles of $CO_2$ also affect the accuracy of estimation for regional carbon fluxes in atmospheric transport model, and can help elucidate the global carbon cycle and climate change. Many experiments have been conducted to measure the vertical profiles of $CO_2$, $CH_4$, and other trace gases. The AirCore sampling system can be used to obtain vertical profiles of $CO_2$ and $CH_4$ from near the surface to 8–12 km with high accuracy (Karion et al., 2010; Membrive et al., 2017). Active remote sensing of atmospheric $X_{CO2}$ with

the Raman lidar (light detection and ranging) technique has been developed and used to measure $CO_2$ vertically in the troposphere (Zhao et al., 2007; Gong et al., 2013; Han et al., 2017). $CO_2$ concentrations were measured at 8–12 km by Tohoku University (Sendai, Japan) through flask sampling on a commercial airliner operated by Japan Airlines (JAL) between Japan and Australia in 1984 and 1985 (Nakazawa et al., 1991). The Comprehensive Observation Network for TRace gases by AIrLiner (CONTRAIL) project installed continuous $CO_2$ measurement equipment onboard aircraft operated by JAL for in situ

measurement (Machida et al., 2008). The data for CONTRAIL are collected at altitudes between a few kilometers and 10 km,

taking advantage of the frequent movement of commercial aircraft around the world. The Civil Aircraft for Remote Sensing and In Situ Measurements Based on the Instrumentation Container Concept (CARIBIC) project (Brenninkmeijer et al., 1999; Brenninkmeijer et al., 2007) aimed to observe trace gases such as CO, $O_3$, and $CO_2$ by deploying measurement equipment in passenger aircraft. The HIAPER Pole-to-Pole Observation (HIPPO) project involved a sequence of five global aircraft

measurement programs to sample the atmosphere from near the North Pole to the coastal waters of Antarctica (Wofsy, 2011). Direct measurements that are independently collected from the aircraft provide validation information for satellite products. Several studies have shown that profile measurements of $CO_2$ and $CH_4$ obtained using aircraft and AirCore are useful for bias correction of both TCCON measurements (Deutscher et al., 2010; Hedelius et al., 2016) and satellite products (Araki et al., 2010; Inoue et al., 2013, 2014; Miyamoto et al., 2013; Frankenberg et al., 2016; Wunch et al., 2017).

Three satellites designed for $CO_2$ measurement, TanSAT (Yang et al., 2018; Yang et al., 2020), GMI/GF-5 (Li et al., 2016), and GAS/FY-3D (Qi et al., 2020), were launched into space in 2016, 2017, and 2018, respectively. Measurement of profiles is crucial to further validating the retrieved hyperspectral measurements from these three satellites. Because the algorithm for satellite retrieval requires *a prior* profiles based on the model and in situ measurements, the lack of direct and independent airborne observations may increase the bias in the satellite results over China.

In this study, in situ aircraft-based measurements of $CO_2$ and $CH_4$ were conducted in Jiansanjiang, Northeast China, in August 2018. An ultraportable greenhouse gas analyzer (UGGA; model 915-0011; Los Gatos Research, San Jose, CA, USA) was used onboard the aircraft to measure the vertical mole fractions of $CO_2$ and $CH_4$ at altitudes of 0.6–7 km. Descriptions of the aircraft and onboard instruments are provided in Section 2. Details of the experimental site and the flight trajectory are provided in Section 3. A comparison of the profiles obtained using aircraft with OCO-2 and the assimilation system Tan-Tracker (v1) is

described in Section 4. Finally, the methods used to calculate $X_{CO2}$ and extrapolate in situ profiles, as well as error estimation, are discussed in Section 5.

## 2 Methods

### 2.1 Aircraft Instrumentation

The aircraft used for this experiment was a Beechcraft King Air 350ER, which is a twin-turboprop aircraft designed for weather

modification missions and measurement of trace gases and aerosols by the China Meteorological Administration (CMA). The cruising speed and maximum speed of the aircraft are 441 and 561 km/h, respectively. Temperature, wind speed, relative humidity, and other meteorological data were detected and recorded by an Aircraft Integrated Meteorological Measurement System (AIMMS-20AG) installed on the aircraft. The geolocation information including latitude, longitude, ambient pressure and height of the aircraft is also measured by AIMMS-20AG. The relative humidity is calculated by temperature and dew

point, measured by the Total Temperature Sensor (Model 102 Type Non-De-iced, Rosemount Aerospace Inc) and Dew Point Hygrometer (Model 137 Vigilant™, EdgeTech), respectively.

The ultraportable greenhouse gas analyzer, UGGA (model 915-0011; Los Gatos Research), was connected to an aircraft-based impactor inlet system which consists of CVI (Model 1204; Brechtel Manufacturing Inc.) and ISO inlet (Model 1200; Brechtel Manufacturing Inc.) in the pressurized cabin for continuous measurement of $CO_2$ and $CH_4$. The CVI and/or ISO inlets were

mounted on the top of the aircraft body as shown in Figure 1, and the air flow rate of the inlets was kept constant by the automatic air flow controller of the inlets (Aircraft-based Counterflow Virtual Impactor Inlet System CVI - Model 1204, Brochure; Isokinetic Inlet System ISO Inlet - Model 1200, Brochure). The UGGA uses a laser absorption technology called off-axis integrated cavity output spectroscopy to determine the trace gas concentration with a high precision of $< 300$ ppb ($CO_2$) and $< 2$ ppb ($CH_4$) and a 10-s response time (UGGA user manual, model 915-0011; Los Gatos Research) and was tested and

controlled in the laboratory. As shown in the in-flight schematic diagram (Figure 1), the external oil-less diaphragm vacuum pump (F-9A 08-03, GAST) was mounted between the CVI inlet and/or the ISO inlet, with the maximum pressure of 31.15 lpm (litter per minute) to keep a stable airflow. The ISO inlet was used as the aircraft pass through clouds, and CVI inlet was used in the other times. Similar system for airborne GHG measurement has been reported by O'Shea et al. (2013) and Palmer et al. (2013).

During the flight, the pressure of the sample cavity was kept constant by a small pump inside the instrument with the airflow about 0.3 lpm. The sample cavity temperature was also kept stable and constant by the temperature controller of the instrument. The instrument automatically recorded and saved the temperature and pressure in the cavity during operation. According to the records, the standard deviation of the cell pressure during three flights is 0.029, 0.029, 0.033 on 7, 9 and 10 August and the range of the cell pressure on each flight is below 0.12 torr. For the cell temperature, the standard deviation is 0.46, 1.55 and

1.18 on each day and the range is below 3.11℃. The UGGA was calibrated against standard GHGs (provided by the National Institute of Metrology, China) before takeoff and after landing of each flight to ensure the accuracy of the data measured with the UGGA. Before this study, the GHG standard gases have been used by the CMA, Chinese Academy of Sciences, and other scientific research institutions for calibration and validation, showing that these standard gases have good performance and reliability. The standard gas we used is based on dry and clean air with greenhouse gases known concentration values, filled

in a 29.5L aluminium alloy cylinder with silanization and other special treatment on the inner wall, traceable to the world meteorological organization global atmospheric observation network (WMO-GAW) level 1 standard gas. The concentration of the $CO_2$ is 400.13 ppm and $CH_4$ is 1.867 ppm. The standard gas we used has been measured in the laboratory for the proportion of $\delta 13C$ in $CO_2$. The range of the proportion is -8.0‰ to -8.2‰, close to the natural content, so it will not cause significant isotopic effect on the measurement of $CO_2$ by optical method and meet the requirements of standard gas (Yao et

al., 2013). Just before taking off, UGGA was calibrated against standard gas, and the stability of instrument was checked and tested again using the same standard gas of $CO_2$ and $CH_4$ immediately after landing. As shown in Figure 2, the concentration of $CO_2$ and $CH_4$ before and after landing is stable around the values of standard gas concentration, and there was almost no drift after the flight. The precision and reparability of the instruments are also checked and tested multiple times in laboratory and the results show that it is stable and good for the measurements.

## 2.2 Tan-Tracker and OCO-2 data

Based on the nonlinear least squares four-dimensional variational data assimilation algorithm (NLS-4DVar) and the Goddard Earth Observing System atmospheric chemistry transport model (GEOS-Chem), Tan-Tracker provides surface flux inversion estimates and profiles of $CO_2$ with 47 levels of vertical resolution from the surface to 0.03 hPa and horizontal resolution of $2.5° \times 2°$. The NLS-4DVar assimilation model Tan-Tracker (v1) and OCO-2 $X_{CO2}$ (v9r) retrievals are used to optimize surface terrestrial ecosystem $CO_2$ flux and ocean $CO_2$ flux, while prior fossil fuel emission and prior fire emission remain unchanged (details of model setting and prior flux information can be found in Han and Tian, 2019).

The Orbiting Carbon Observatory-2 (OCO-2), successfully launched on 2 July 2014, obtained global measurement of CO2 through hyperspectral measurement of reflected sun light from earth atmosphere in one NIR and two SWIR bands centre at 0.76, 1.61 and 2.06 µm, more details about the mission, retrieving algorithm and data characteristic can be found in Crisp et al. (2008) and O'Dell et al. (2012). The uncertainty and bias of the $X_{CO2}$ products related to surface properties, aerosol and cloud, and the retrieving algorithm has been reported by Butz et al. (2009), Jung et al. (2016) and Connor et al. (2016). The OCO-2 data (V9r) including $X_{CO2}$, $CO_2$ profile and the a priori profile were used in this study.

## 3 Experimental Site

Aircraft measurement were carried out from 7 to 10 August over Jiansanjiang (47.11°N, 132.66°E, 61 m above sea level), located in Heilongjiang province, Northeast China. Figure 3 shows the geolocation of the Jiansanjiang aircraft and the flight paths. The area is mostly covered with large tracts of farmland. Rice cultivation is carried out primarily in summer, and crop growth is vigorous during this period. Due to the influence of plant photosynthesis, a large amount of $CO_2$ uptake occurs near the surface.

Three profiles were obtained between around 08:00 and 11:00 in local time (GMT+8) on 7, 9 and 10 August, 2018. The aircraft is designed for weather modification by China Meteorological Administration (CMA), so the infrastructure of the aircraft and the gas flow system are also designed and completed in the USA by the team of weather modification agency. CMA is in charge of the flight route, and there is a chance (several times later are planning) that it can carry the greenhouse gas analyzer to measure the profiles of $CO_2$ and $CH_4$. The greenhouse gas analyzer was loaded on the aircraft and some parts of air flow arrangements were modified to better fit the requirement for greenhouse profile measurement. Due to the logistical problem and the ATC restriction, we must fly in the morning from around 7:30 to 11:00 (local time) of these days to avoid obstructing civil aviation. The details of the three flights are listed in Table 1.

The flight trajectory on 7 August is shown in Figure 4. The aircraft climbed up quickly and directly to the maximum height at about 7.5 km 30 min after taking off, and then descended down step by step at about every 300 m. Since the 3-D Figure in these three days looks identical, the flight trajectory of the other two days (9 and 10 August) is not shown in Figure 4. Considering the sensitivity of the UGGA response, measurements during the ascent were discarded due to the rapid changes in air pressure, and only data collected while spiralling downward were regarded as valid and analyzed further. Data recorded

below 0.6 km were also rejected because samples were easily contaminated with exhaust emissions during the slowing and descent of the aircraft before landing. The spiral descent of the aircraft lasted about 2.5 h on each of the three sampling days, and the number of effective layers of measurements are 17, 21 and 20, respectively, in 7, 9 and 10 August..

## 4 Data Processing

### 4.1 Water vapor correction

The mole fraction of $CO_2$ or $CH_4$ measured during flight is the volume in proportion to the air containing water vapor, which cannot be directly compared with values from other data sources due to different water vapor contents of the sampled air. Therefore, the effect of water vapor is corrected and the mole fractions of $CO_2$ and $CH_4$ to dry air are given by:

$$f_{gas\_dry} = \frac{f_{gas} \cdot p}{p - p_{H_2O}} \tag{1}$$

where $f_{gas\_dry}$ (mol/mol) is the mole fraction of a gas in dry air, and $f_{gas}$ (mol/mol) is the measured mole fraction of a gas under the real air conditions with water vapor. $P_{H_2O}$ is water vapor pressure in hPa, which can be calculated as:

$$p_{H_2O} = e_s \cdot RH \tag{2}$$

where $e_s$ (hPa) is the saturated water vapor pressure at temperature T (K) at aircraft altitudes, which can be derived from the Clausius–Clapeyron equation:

$$\ln \frac{e_s}{6.11} = \frac{L_v M_w}{R}(\frac{1}{273} - \frac{1}{T}) \approx 5.42 \times 10^3 (\frac{1}{273} - \frac{1}{T}). \tag{3}$$

where $L_v = 2.500 \times 10^6$ J Kg$^{-1}$, $M_w = 18.016$, which is the molecular weight of water, R = 8.3145 J K$^{-1}$mol$^{-1}$, and $e_s$ (in hPa) at temperature T (in K). Pressure p (hPa) of the ambient atmosphere are measured by the aircraft meteorology system, AIMMS-20AG, and the temperature T (K) was measured by Total Temperature Sensor (Model 102 Type Non-De-iced). The relative humidity RH (%) was calculated by the dew point and temperature. The dew point data is obtained by Dew Point Hygrometer (Model 137 Vigilant™, EdgeTech).

### 4.2 Accuracy and precision

Before aircraft takeoff, the clocks of the UGGA, AIMMS-20AG, the Total Temperature Sensor and other instruments were adjusted to match those of the $CO_2$, $CH_4$, and weather system measurements, synchronizing these data to the altitude and geolocation of the aircraft. The data from UGGA and synchronous meteorology measurements, including temperature, pressure, and humidity of ambient atmosphere, are recorded every second, then smoothed with a 10-s running average to further remove errors caused by temporal mismatch considering the response time of the UGGA. Because the flights followed the spiral trajectories that descended approximately every 300 m, only data collected during level flight were retained and analyzed, whereas data from the descent periods were removed to avoid the effects from vertical variations in sampling during

rapid descent. The time points at the beginning and end of level flight are determined according to the altitude and its variation of the aircraft. Considering the residual time of the GHG measurement system, the data obtained 220s from the start of the level flight is considered to be observed when the aircraft is descending rather than in level, which may cause uncertainty of the measurement. Therefore, the data were kept after the level flight starting for 220s. If the duration time of certain level flight lasted less than 220s, the data observed during that level flight were also discarded.

The instrument was calibrated against the standard gas before and after each flight. All of the measurements during the calibration process, including the standard gas used for calibration can trace back to WMO scale. The maximum and the average values of the difference between the standard gas and the measurement of the instrument of each day was considered as the accuracy of the aircraft data. For the precision, noted that the instrument was not continuously calibrated against the standard gas during the flight, we calculated the one standard deviation of the data in each level flight, and the maximum of the average value of 1-σ on each day is considered as the precision of the aircraft measurement. The accuracy of $CO_2$ and $CH_4$ is below 0.66 ppm and 0.002 ppm, 0.16% and 0.10% of the $CO_2$ and $CH_4$ concentration in standard gas, respectively. For precision, the 1-σ value is below 0.71 ppm and 0.0062 ppm for $CO_2$ and $CH_4$, respectively.

## 5 Results and Discussion

### 5.1 $CO_2$ and $CH_4$ profiles

Figure 5 shows vertical profiles of the $CO_2$ and $CH_4$ mole fractions measured with the UGGA during the flight over Jiansanjiang, which is an agricultural area that produces a large amount of rice. The $CO_2$ concentration increased with height in the troposphere (Figure 5a), which may resulted from $CO_2$ uptake by rice plants near the surface during the summer growth season. The greater increase rate of $CO_2$ in the lower troposphere on 7 August compared to the other two days was probably attributed to differing weather conditions on the three sampling days. 7 August was a sunny day, but it was overcast on 9 and 10, August, which may have weakened photosynthesis in rice and reduced $CO_2$ uptake. During all three flights, the maximum mole fraction of $CO_2$ reached about 418 ppm in the free troposphere at the top of the profile.

The mole fraction of $CH_4$ (Figure 5b) showed a consistent decrease in concentration with height, ranging from 1.95 to 2.10 ppm from about 2 km to near the surface, possibly as a result of $CH_4$ emissions from agricultural activity at the surface. $CH_4$ showed low variability of less than 0.5 ppm at higher altitudes, from above 2 to 7 km, indicating a well-mixed vertical structure of $CH_4$ in the free troposphere.

Comparing the $CO_2$ and $CH_4$ observation data, the mole fraction of $CH_4$ varied less than that of $CO_2$ from 1.5–2 km up to the free troposphere, with a stable value of about 1.925 ppm. This stability indicated that $CH_4$ was evenly mixed at these heights and there were no obvious sources or sinks of $CH_4$. Conversely, $CO_2$ rose with increasing altitude in the free troposphere from about 400 to 418 ppm. This increase may have been due to photosynthesis by vegetation and the large number of crops planted locally, creating a $CO_2$ sink at the surface, and causing the $CO_2$ concentration to rise with height in the free troposphere. The

results show that the vertical profile of $CO_2$ in summer increases with height in the upper troposphere, whereas that of $CH_4$ changed little with height and was relatively stable over Jiansanjiang area during experiment.

## 5.2 Comparison of profiles from the model and satellite product

Aircraft measurements were compared with $CO_2$ data obtained from OCO-2 (v9r) retrievals and the recently developed data assimilation system for the global carbon cycle, Tan-Tracker (v1) (Han and Tian, 2019). The assimilation data are collected and linearly interpolated spatially and temporally based on the geolocation of the observation site and time. Because no data were obtained from OCO-2 (v9r) over Jiansanjiang during the flight, the results of OCO-2 within $1° \times 1°$ spatially at the closest time to the flight were used for comparison, which were collected on 5 August. The height of the profile is available on the satellite product.

The structure of $CO_2$ varying with height could be roughly divided into three segments: surface to 2 km, 2 to 3 km, and 3 to 8 km (Figure 6). Below 2 km, $CO_2$ of Tan-tracker model is assumed to be well mixed and uniformly distributed with height, with values ranging from 385 to 395 ppm, therefore, the model could not reproduce the strong decrease in $CO_2$ from 2 km to the surface due to uptake by vegetation. From 2 to 3 km, $CO_2$ increased to about 400 ppm with altitude. The averaged satellite retrieval profiles correctly reproduced the decrease in $CO_2$ from 2 km to the surface, but the decrease rate was lower than those of in situ profiles, decreasing from 393 ppm at 2 km to 390 ppm near the surface. Flight data showed a significant $CO_2$ sink in this region, most notably on 7 August when it decreased from 400 ppm at 2 km to 380 ppm at 0.6 km. The impact of ground sinks was more pronounced and apparent than the results from satellite inversion and model simulations, indicating that the strong variations in the lower atmosphere and planetary boundary layer (PBL) should be more carefully considered in model and retrieval algorithms. Between 2 and 4 km, aircraft profiles showed a relatively uniform mixing level of $CO_2$, with roughly stable concentrations around 400 ppm.

In general, all profiles from aircraft, satellite retrieval, and model showed a similar vertical distribution trend in the troposphere above 2 km, but with differences in values. The average of the difference between OCO-2 and the aircraft profiles above 2 km is 4.22 ppm, 8.16 ppm and -2.57 ppm in 7, 9 and 10 August, respectively. The volume mole fraction of $CO_2$ from both satellite and aircraft measurements indicated a $CO_2$ sink. GHGs profiles have rarely been observed before near the experiment site, or over Northeast of China as far as we know. The model simulations are based on data of regional emission inventory. The accuracy of simulated profiles and concentrations near surface over the experiment site still remains unknown. So continuous and regular observation of the GHGs profiles are necessary for better understanding of the regional emission amounts and the variation of the GHGs.

## 5.3 Comparison of X$_{CO2}$ products

The total column amount of $CO_2$, can be derived by integrating the $CO_2$ concentrations from the surface to the top of the atmosphere under the assumption of hydrostatic conditions:

$$VC_{CO_2} = \int_0^{P_s} \frac{f_{co_2}^{dry} \cdot (1 - f_{H_2O})}{g(p) \cdot m(p)} dp \tag{4}$$

$$m = [m_{H_2O} \cdot f_{H_2O} + m_{air}^{dry} \cdot (1 - f_{H_2O})]$$

where $VC_{CO_2}$ is the total column amount of $CO_2$, $f_{co_2}^{dry}$ is the dry-air mole fraction (DMF) of $CO_2$ (mol mol$^{-1}$), $f_{H_2O}(p)$ is the aircraft profile of $H_2O$ (mol mol$^{-1}$), which is measured by the onboard AIMMS system, $m(p)$ is the mean molecular mass of wet air, and $g(p)$ is gravitational acceleration, $m_{H_2O} = 18.02 \times 10^{-3}/N_A$ kg/molecule, $m_{air}^{dry} = 28.964 \times 10^{-3}/N_A$ kg/molecule, and $N_A$ is Avogadro's constant. Data beyond the flight limits are taken from National Centers for Environmental Prediction (NCEP) reanalysis data interpolated to the time of flight.

The column-averaged DMF of $CO_2$ ($X_{CO2}$) from aircraft measurements was calculated based on the method of Wunch et al. (2010), which considers the average kernel in OCO-2 satellite retrievals:

$$X_{CO_2}^{in\ situ} = X_{CO_2}^{a} + \sum_j h_j a_j (t_{in\ situ} - t_a)_j \tag{5}$$

where $a$ is the average kernel (Rodgers and Connor, 2003), $X_{CO_2}^{a}$ is the column-averaged DMF for the a priori profile $t_a$, $h_j$ is the pressure weighting function of OCO-2, and $t_{in\ situ}$ is the in situ profile from aircraft measurement.

Because in situ measurements available from aircraft are limited, values outside the aircraft's vertical observation range must be estimated to calculate $X_{CO2}$. Two extrapolation methods were used to extend the profile of the aircraft measurements and then estimates the $X_{CO2}$ value of the in-situ measurement respectively. 1) The unknown part of the aircraft profile was directly from the OCO-2 a prior profile. 2) A well-mixed and constant mixing ratio of $CO_2$ is assumed from the surface to the lower limit of flight, and from the upper limit of flight to the tropopause. The $CO_2$ concentrations above the tropopause were calculated with an empirical model (Toon and Wunch, 2014) which considers tropopause height as well as realistic latitude and time dependencies through curve fitting of data from high-altitude balloons, AirCore, Observations of the Middle Stratosphere balloon, and aircraft. In general, the mole fraction of $CO_2$ decreased exponentially with height from the tropopause to upper stratosphere, and the tropopause height was obtained from NCEP reanalysis data with a $2.5° \times 2.5°$ resolution, which was linearly interpolated to the geographic coordinates of Jiansanjiang. Figure 7 shows the extrapolated $CO_2$ profiles using method (2).

$X_{CO2}$ calculated from the aircraft measurements and differences with that from OCO-2 are listed in Table 2 and 3. The results showed that $X_{CO2}$ values of OCO-2 were lower, with an average difference of $-4.68 \pm 0.44$ ppm ($-1.18\% \pm 0.11\%$) and $-5.09 \pm 1.28$ ppm ($-1.28\% \pm 0.32\%$) by method (1) and by method (2).

Uncertainties induced by extrapolation of profiles outside the height limits of aircraft and errors in tropopause estimation were analyzed. Errors in extrapolation of the profile below the lower limit and above the upper limit of flight were estimated by recalculating $X_{CO2}$ after a 1-ppm positive shift in the $CO_2$ concentrations at these altitudes. For method (1), since the values of $CO_2$ mole factions of unknown part is the same as that of the OCO-2 a priori profile, as Eq. (5) shows, no extra uncertainty

would be introduced by extrapolation. But for method (2), as the profile is assumed to decrease exponentially with height above the tropopause, the height of the tropopause also introduces uncertainties for $X_{CO2}$. Table 4 lists the errors resulting from three sources: 1) uncertainties from in situ measurement, 2) extrapolation of the profile in the PBL where no in situ measurements were collected, and 3) profile assumptions above the upper limit of flight observations. Errors due to uncertainty in tropopause height were analyzed by shifting the tropopause height upward by 1 km, and the results are also listed in Table 4. These results indicated that the extrapolation method and assumptions used to construct profiles where no measurements were made were the primary source of errors, among which the greatest error was from the profile above the upper limit of the flight (0.323 ppm). Errors due to uncertainty in the tropopause height were also non-negligible. Because of the lack of observation data near the surface, the missing measurements were directly replaced by the data at the lowest altitude measured by the aircraft. The error caused by this practice is shown in Table 4, with an average of 0.079 ppm for $X_{CO2}$. This is also the impact of the lack of near-surface observations on the overall $X_{CO2}$ estimation. Therefore, observations from near the surface to about 1 km from other method, such as in-situ GHG measurement by tethered balloon and high tower, are necessary for accurate estimation of $X_{CO2}$.

**6 Conclusion**

The vertical distributions of $CO_2$ and $CH_4$ were measured by using a Beechcraft King Air 350ER over Jiansanjiang, an extensive paddy area in Northeast China, and three vertical profiles from 0.6 to 7.5 km were obtained on 7, 9 and 10 August. Measurements of the mole fraction of $CO_2$ showed an increase with height, whereas $CH_4$ decreased with height. These results are reasonable, because paddies are sinks for $CO_2$ and sources of $CH_4$ during the summer growing season. Comparing the observed profiles from aircraft with those from the carbon cycle data assimilation system Tan-Tracker (v1) and OCO-2 retrievals showed that the general vertical structure was consistent, but the values of mole fraction of $CO_2$ from Tan-Tracker and OCO-2 had negative bias estimates. The average bias between aircraft and OCO-2 is $-4.68 \pm 0.44$ ppm ($-1.18\% \pm 0.11\%$). The uncertainty mainly resulted from extrapolation of the profile beyond the flight limit, where no in situ measurements were available.

**Data availability**

Data used in this study are available from the corresponding author upon request (dmz@mail.iap.ac.cn).

**Author contributions**

M. Duan and X. Sun determined the main goal of this study. X. Sun carried it out, analyzed the data, and prepared the paper with contributions from all co-authors. Y. Gao and N. Chen provided technical guidance for related instrument.

**Competing interests**

The authors declare that they have no conflict of interests.

**Acknowledgements**

This work is supported by the National Natural Science Foundation of China (No. 41527806 and No.41705014). We also acknowledge numerous staff of the Weather Modification Center, China Meteorological Administration for supporting the experiment and the instrumentation onboard the aircraft. We also thank Jiansanjiang Airport for providing the experimental

site and arranging time for conducting the campaign.

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

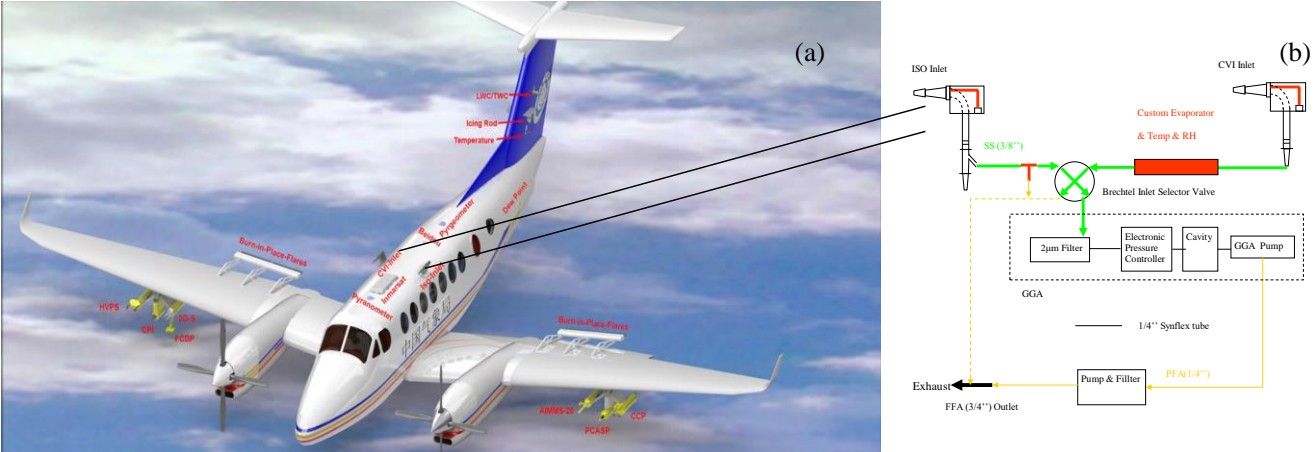

**Figure 1. (a) The outside view of the Beechcraft King Air 350ER instrumentation. (b) The schematic diagram of the greenhouse gases sample airflow.**


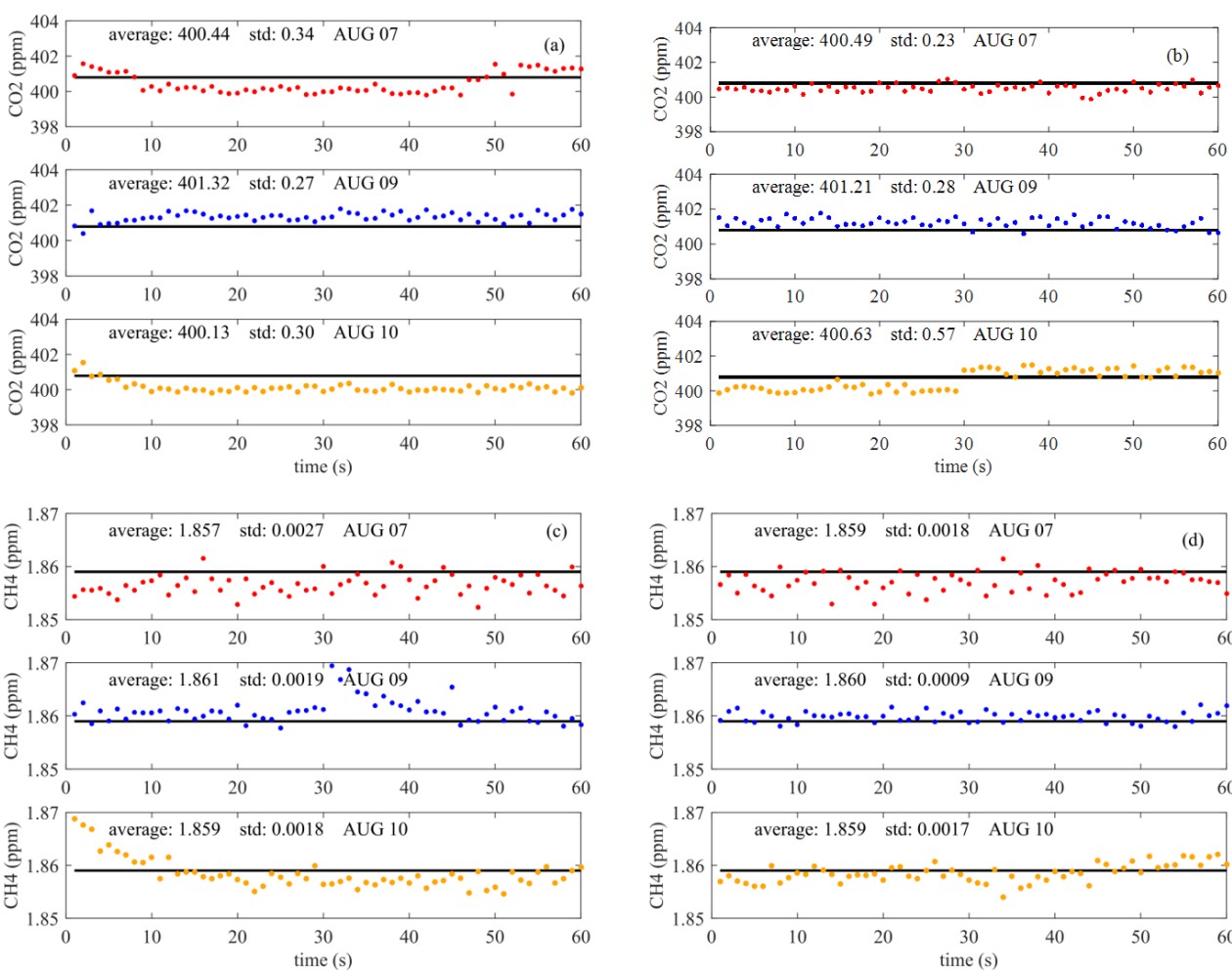

**Figure 2. The concentrations of CO₂ (a) and CH₄ (b) before the flight, and the concentrations of CO₂ (a) and CH₄ (b) after the flight obtained during the calibration, with the values of standard deviation and average of each calibration.**


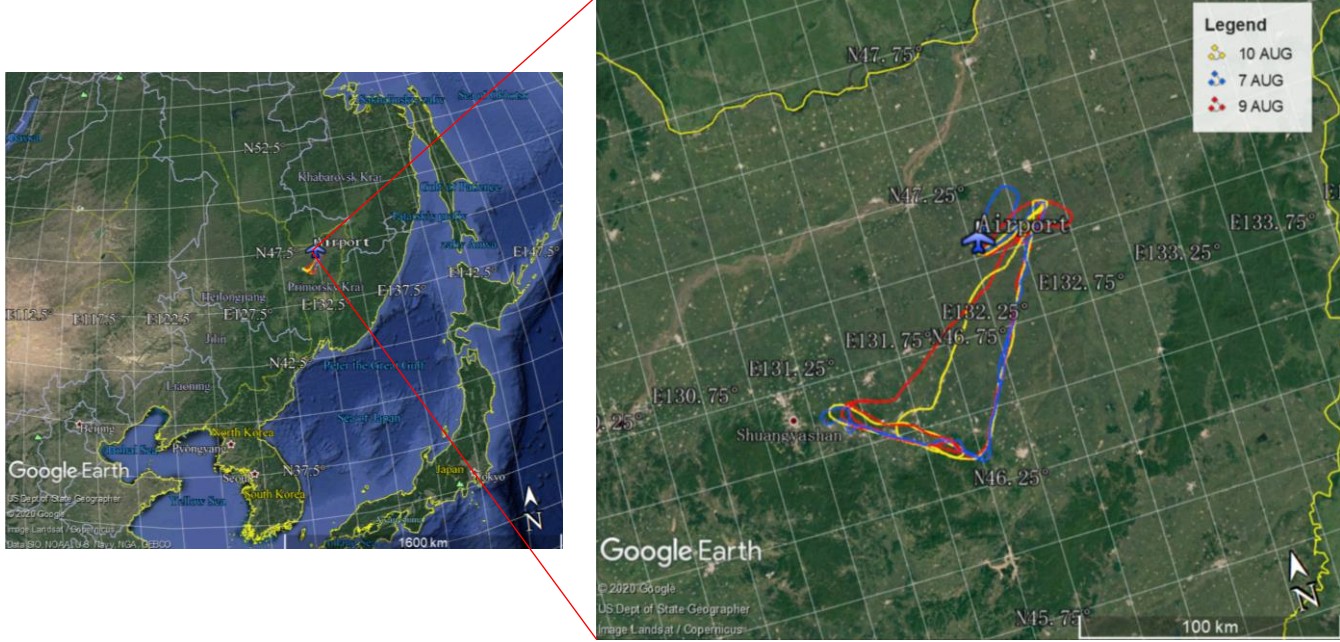

**Figure 3.** Observation area for aircraft-based measurement of $CO_2$ and $CH_4$ over Jiansanjiang, Northeast China, and the flight paths on 7, 9, 10 August.


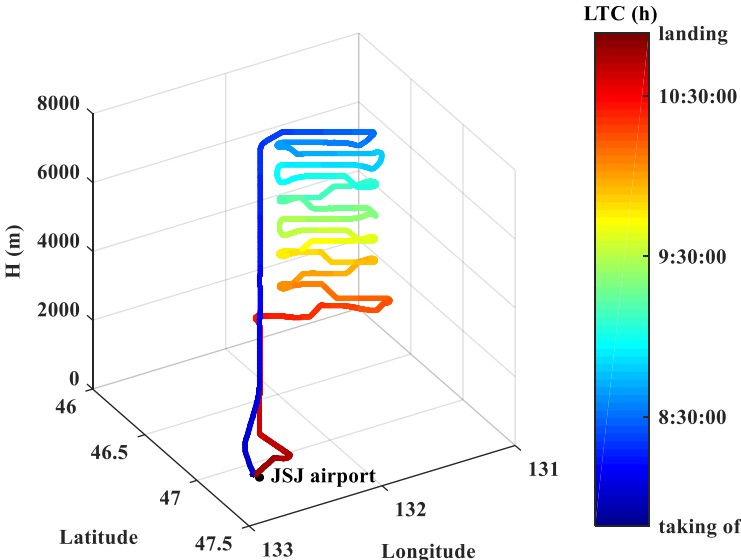

**Figure 4. Trajectory on the 7 August, 2018 in Jiansanjiang. The color scale shows the progression of time (LTC), where blue represents the start time of the data profile, and red represents the end time.**

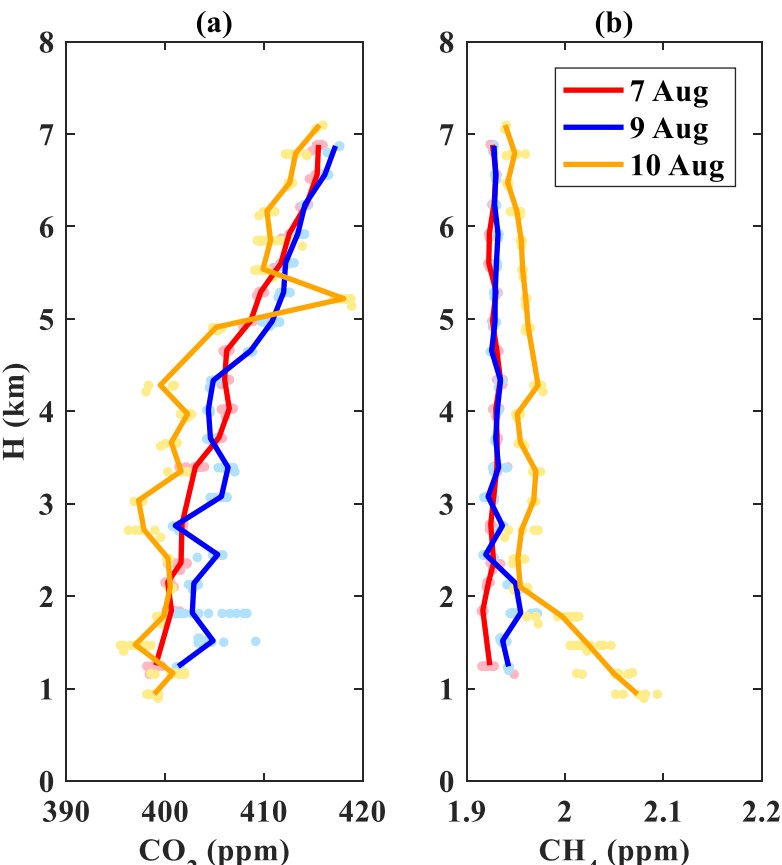

**Figure 5 Vertical profiles of (a) CO₂ and (b) CH₄ observed on 7 (blue), 9 (red), and 10 (yellow), August, 2018, over Jiansanjiang measured in situ with aircraft. The aircraft-based in situ measurement data are indicated with dots, and averaged data for each flat flight stage are shown as lines.**

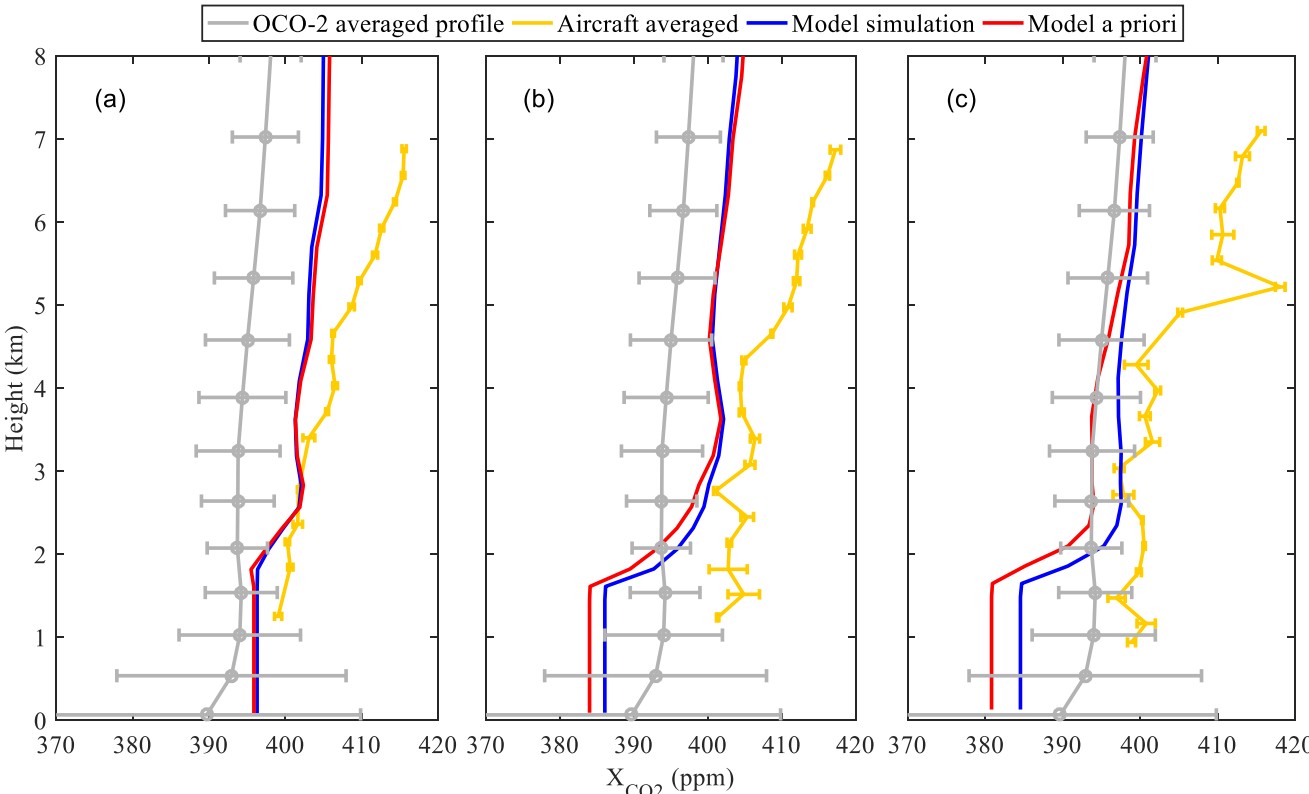

**Figure 6. Comparison of aircraft measurements (in situ measurement data are shown by the yellow line) with 1 standard deviation (yellow bars) collected on (a) 7, (b) 9, and (c) 10, August Tan-Tracker (v1) data (blue line) and the a priori profile of it (red line) at the location of Jiansanjiang linearly interpolated to the observation times on (a) 7, (b) 9, and (c) 10 August, and the OCO-2 averaged profile (gray line) for the aircraft flight area from   5 August with 1 standard deviation (grey bars).**

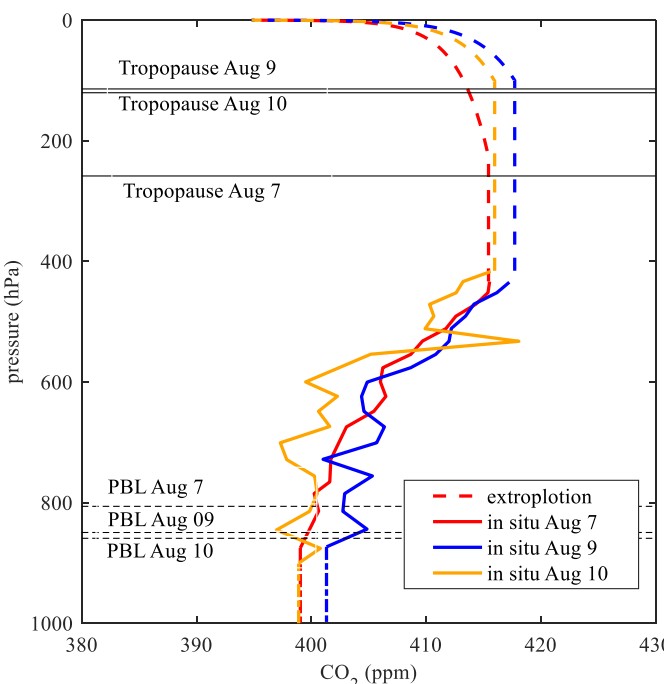

**Figure 7. Extrapolated $CO_2$ profiles observed on 7, 9 and 10 August, 2018, over Jiansanjiang by method (2). Red, blue, and yellow solid lines show the aircraft-based (in situ) data collected on 7, 9 and 10 August, respectively, averaged for each flat stage of the flight. Dotted lines show the extrapolated parts of the profiles, with colors corresponding to sampling dates in accordance with the solid lines. Black horizontal lines show the tropopause height from NCEP reanalysis data.**

**Table 1 Details of the flight on each day.**

| Date | Flight Time (LTC) | Flight Altitude (m) |
|---|---|---|
| 7 August, 2018 | 07:49:08-10:53:32 | 59-7205 |
| 9 August, 2018 | 07:50:19-10:45:57 | 61-7190 |
| 10 August, 2018 | 07:56:02-10:54:11 | 65-7104 |

**Table 2. $X_{CO_2}$ derived from aircraft on each observation day (7, 9 and 10 August) supplemented the aircraft profile by method (1). OCO-2 (V9r) $X_{CO_2}$ were from 5 August, which was the closest time point of $X_{CO_2}$ data from OCO-2 over Jiansanjiang to the observation period. Differences between aircraft $X_{CO_2}$ and OCO-2 are shown in the fourth (ppm) and fifth (%) columns. The average difference and standard deviation are shown in the fifth row.**

| Date | Aircraft* (ppm) | OCO-2 (ppm) | Difference (ppm) | $\dfrac{\text{OCO-2-Aircraft}}{\text{Aircraft}}\cdot 100\%$ (%) |
|---|---|---|---|---|
| 7 August, 2018 | 401.95 | 396.91 | -5.04 | -1.27 |
| 9 August, 2018 | 401.72 | | -4.81 | -1.21 |

| Date | | | -4.19 | -1.06 |
|------|--|--|-------|-------|
| 10 August, 2018 | 401.10 | | | |
| | | Average (1σ) | -4.68(0.44) | -1.18(0.11) |

*The effect of the average kernel was taken into consideration for OCO-2.

**Table 3 The same as Table 2, but for method (2).**

| Date | Aircraft* (ppm) | OCO-2 (ppm) | Difference (ppm) | $\frac{OCO\text{-}2\text{-}Aircraft}{Aircraft} \cdot 100\%$ (%) |
|------|-----------------|-------------|------------------|--------|
| 7 August, 2018 | 401.54 | | -4.63 | −1.16 |
| 9 August, 2018 | 403.45 | 396.91 | −6.54 | −1.64 |
| 10 August, 2018 | 401.02 | | −4.11 | −1.03 |
| | | Average (1σ) | −5.09 (1.28) | −1.28 (0.32) |

*The effect of the average kernel was taken into consideration for OCO-2.

**Table 4. Aircraft integration error budget of $X_{CO_2}$ estimation for method (2). Errors in the three profiles from multiple error sources contributed to the calculation results of the integrated total column. There are four sources of error, similar to previously described error budgets (Wunch et al., 2017): the contribution from the aircraft profile itself, the contribution from the unknown surface to the bottom of the profile, the contribution from the upper troposphere and stratosphere, and error from the tropopause height.**

| Date | PBL errors (ppm) | Upper troposphere and stratosphere errors (ppm) | Tropopause height errors (ppm) |
|------|------------------|-------------------------------------------------|--------------------------------|
| 7 August, 2018 | 0.086 | 0.323 | 0.054 |
| 9 August, 2018 | 0.076 | 0.303 | 0.017 |
| 10 August, 2018 | 0.077 | 0.077 | 0.017 |
| Average | 0.079 | 0.234 | 0.029 |