# Peer review of "In situ measurement of CO2 and CH4 from aircraft over northeast China and comparison with OCO-2 data"

_Atmospheric Measurement Techniques, 2019_

## Referee Comment (RC1) · Anonymous Referee #1 · 29 Dec 2019

Review of "In situ measurement of CO2 and CH4 from aircraft over northeast China and comparison with OCO-2 data" by Xiaoyu Sun et al.

General comments:

The paper by Sun et al. reports from airborne in-situ measurements over North-East China in August 2018. The in-situ profiles derived on three different days are compared with profiles from OCO-2 and a carbon cycle data assimilation data system (Tan-tracker). The topic of the manuscript is of high importance since high-quality observations are needed to enable a better analysis of the global carbon cycle. Specifically, in-situ measurements are highly valuable to study local phenomena in detail and to allow for an evaluation of satellite products. This is especially true and important for regions were observations are rare and the variability of the atmospheric greenhouses are not well constrained, because emission amounts are not well known. Therefore I strongly encourage the authors to continue their work in this field because the gained data sets are highly valuable to the carbon community. However, the manuscript lacks on a detailed description and discussion to support the conclusions drawn by the authors. Personally, I also doubt the quality of the aircraft-borne in-situ measurements and therefore suggest publication of the manuscript only after my main (specific) comments are carefully addressed.

Specific comments:

I suggest to re-structure the manuscript and to expand the section "instrumentation" to "methods" by including a subsection on Tan-Tracker and OCO-2 (including a thorough description of the model products and the derivation of the OCO-2 data product).

Page 2, L41: all-weather?

P3, L92: Which one? AIMMS-20?

P3, L94: Why did you use a CVI inlet? Where there other measurement (aerosol) systems onboard? Please be also more specific w.r.t. to the airborne set-up. Did you need to use an external pump to achieve the large gas flow? How long was the inlet (from the tip to the cell)? I am not aware of a publication which reports the airborne deployment of this kind of analyzer, so I suggest to include a schematic which shows the set-up and the periphery to control cell/inlet pressure, temperature and volume or mass flows.

P3, L96: SL/min? Which kind of Mass Flow controller?

P3, L97: The given values are from the manufacturer and might be valid for controlled laboratory conditions. Usually, the performance on a mobile platform is highly affected by variations of pressure, temperature and/or mechanical vibrations. I assume that this specific instrument is even more sensitive since it is not especially designed for use aboard research aircraft. Did you cross-check the theoretical precision values yourself in the laboratory, e.g. by supplying the system with sample gas of constant $CO_2$ and $CH_4$ mixing ratios? Did you check the sensitivity of your instrument to changes in pressure and temperature? Did you check the short and long-term drift of your instrument´s sensitivity (i.e. over one flight and over a couple of days, respectively)? Did you check the repeatability of your measurements?

P3, L98: What do you mean with response time in this case? Is this the response time of the system to a change in atmospheric concentrations (due to e.g. the residence time in the inlet)? Is it the averaging time to achieve the given precision (in theory)? O is it the flush time of the cell and thus, gives the best achievable time resolution?

P3, L98: Please specify: Where was the pressure controller installed? I assume in front of the instrument? How constant was the pressure during the flight?

P3, L99: Which temperature? The cell temperature? A range of more than 6 degrees seems very huge to me and should impact the sensitivity of the instrument. Did you check this in the lab (see also above)?

P3, L99: Please provide more details on the Standards. How many standards did you use? At which concentrations? How did you calibrate the system? Did you (or some of the other institutions cross-calibrate the standards in a way that they are traceable to the typically used NIST standards?

How stable was the system? How reproducible were the standard measuremenst before and after the fligh?

P4, L107, Figure 1: What do you want to show with this figure? I suggest to zoom in and to include at least the flight patterns of all 3 flights conducted in August 2017. You might also show a series of three figures with all three flight paths plotted over a weather map.

Section 3:

This section is especially weak. I suggest to include information about the flight strategy, as well.

So e.g., why did you fly in in the morning hours (during which the boundary layer develops)? Did you try to match the time of an OCO-2 overflight? Or was this due to logistical (ATC) reasons? Why did you choose to fly over a horizontal distance of 150 km (the swath is a couple of km´s only)? Did you adjust the flight track to measure along-track of OCO-2? Or is this Did you always follow the same flight strategy on the three days? What was the descending rate and the corresponding pressure variation during spiraling down?

P4, L110, Figure 2: From this figure it looks like that you did ~7 constant flight legs, is that correct? I don´t think you need this figure if you provide a horizontal map of the flight patterns as suggested for figure 1, which gives an idea about the flight dimensions in Lat/Lon direction. Instead, I suggest to include a simple time-series of in-situ measured CO2, CH4, and flight altitude for this particular flight.

Section 4: Please keep in mind that there are several ways how variable water vapor levels influence the CO2/CH4 measurements: 1) The dilution effect, 2) variation in the line broadening of the carbon dioxide and methane lines due to varying water vapor concentration, and 3) nonlinearity of the reported water vapor concentration due to self-broadening of the water vapor line.
Here you discuss the dilution effect which certainly is the most important one. However, the water concentration measurement must be highly accurate to allow for a meaningful accuracy in the derived dry gas concentrations. Therefore I´d like to see an in-depth error analysis for the approach used herein. Moreover, I suggest to include the time-series of measured relative humidity and derived water vapor (including error bars!), at least in the supplement.

P5, L 132: All data are recorded at 1s and then smoothed to remove errors because of the response time? As mentioned above, please be clear in the use of your wording w.r.t. response time. The residence time usually can be corrected for if e.g. volume flow and inlet pressure are known.

Section 5:

Figure 3: This figure shows that you did much more constant flight legs than it seems from figure 2. Do these dots represent 10s values? What is real variability and what is instrument precision? Variation in CO2 on each leg is large (maybe also because of the large horizontal distance), the vertical variability on Aug 10 seems larger than the horizontal variability on that day. Is this a real atmospheric feature and do you have any explanation for this? Also, the boundary layer variability on 9 Aug seems much larger than on other days. Is this a horizontal gradient? Please include a more in-depth discussion on these profiles. I´d like to also see the standard deviation or even better, median instead of average values including some percentiles. What was the boundary layer height? Please include also a vertical profile of met. variables, at least in the supplement.

P5, L 142: "was attributed to different weather conditions": Please provide more details on this hypothesis.

P6, L161: Please give a short introduction about the Model, the used a-priori information and the simulations – and the difference. Which data are assimilated? The OCO-2 data? Doesn´t look like. The aircraft data?

P6, L 163: "The variation of CO2 ….". I don´t understand this sentence. Do you talk about the aircraft measurements? What about uncertainty bars for the aircraft data?

P6, L165: Reproducing CO2 uptake from vegetation by a model is highly challenging, but I do not see any information from the model (neither a-priori or simulated). Is this what you mean with "Below 2km, CO2 is assumed to be vertically mixed…"?

P6, L166: OCO-2 data were averaged over what area and what time? Please provide a graphical explanation which OCO2-Data you used. How did you get the vertical information?

P6, L175: "…with large differences in values". So do you have any explanation? Apart from the quality of the in-situ data, one reason might be the comparison of measurements on different days. To get an impression about the day-to-day variability in that region, you might have a look at a longer time-series of OCO-2 data. You also might have a look at the weather conditions (low or high pressure systems, frontal crossings) and how these may have influenced the day-to-day variability.

P7, L194: Did you use all these observations (aircore, balloon, aircraft) for your specific case or is this a general description? Is this Tan-Tracker? Please be more specific.

P7, L197: This information come much too late (should be at the beginning of section 5.2).

P7, L199: You compare XCO2 values with the in-situ measured data. The variability of the latter is nearly 40 ppm, which is not at all captured by the OCO-2 average profile. In my opinion, you can´t compare column values and derive a bias (especially not with the accuracy given).

Table 2: Please provide details on the uncertainty analysis for the aircraft errors: accuracy (traceability to WMO scale) and precision.

---

## Author Comment (AC1) · 14 Mar 2020

Response to Referee comment 1

The authors thank all reviewers for their constructive comments and suggestions, which have helped us to improve the quality of this paper both in sciences and writing. All comments are carefully considered and responded. The response in blue italic letters follow each comments in black.

1. General comments: The paper by Sun et al. reports from airborne in-situ measurements over North-East China in August 2018. The in-situ profiles derived on three

different days are compared with profiles from OCO-2 and a carbon cycle data assimilation data system (Tan-tracker). The topic of the manuscript is of high importance since high-quality observations are needed to enable a better analysis of the global carbon cycle. Specifically, in-situ measurements are highly valuable to study local phenomena in detail and to allow for an evaluation of satellite products. This is especially true and important for regions were observations are rare and the variability of the atmospheric greenhouses are not well constrained, because emission amounts are not well known. Therefore I strongly encourage the authors to continue their work in this field because the gained data sets are highly valuable to the carbon community. However, the manuscript lacks on a detailed description and discussion to support the conclusions drawn by the authors. Personally, I also doubt the quality of the aircraft borne in-situ measurements and therefore suggest publication of the manuscript only after my main (specific) comments are carefully addressed. Specific comments: I suggest to re-structure the manuscript and to expand the section "instrumentation" to "methods" by including a subsection on Tan-Tracker and OCO-2 (including a thorough description of the model products and the derivation of the OCO-2 data product).

Thank you so much for the advice on this study. The section 2 "Instruments" is rewritten as "Methods" and the original context of section 2 changed to 2.1 "Aircraft Instrumentation". In addition, we added section 2.2 "Tan-Tracker and OCO-2 data" to describe the Tan-Tracker (v1) model and the OCO-2 data used in this article, please see detail in section 2.2, Page 5, Line 132 in the revised manuscript: "Based on the nonlinear least squares four-dimensional variational data assimilation algorithm (NLS-4DVar) and the Goddard Earth Observing System atmospheric chemistry transport model (GEOS-Chem), Tan-Tracker provides surface flux inversion estimates and profiles of $CO_2$ with 47 levels of vertical resolution from the surface to 0.03 hPa and horizontal resolution of $2.5° \times 2°$. The NLS-4DVar assimilation model Tan-Tracker (v1) and OCO-2 $XCO_2$ (v9r) retrievals are used to optimize surface terrestrial ecosystem $CO_2$ flux and ocean $CO_2$ flux, while prior Fossil Fuel emission and prior Fire emission remain unchanged (details of model setting and prior flux information can be found in Han and Tian, 2019).

The Orbiting Carbon Observatory-2 (OCO-2), successfully launched on 2 July 2014, obtained global measurement of CO2 since September 2014.Three bands at 0.76, 1.61 and 2.06 $\mu$m was used, and spectrometers of OCO-2 measure high-resolution near-infrared reflected sunlight from Earth's surface. Global XCO2 is the main product of OCO-2 with high precision, more details about the mission, the retrieving algorithm and data characteristic is expected to be found in Crisp et al. (2008) and O'Dell et al. (2012). The uncertainty and bias of the XCO2 products related to surface properties, aerosol and cloud, and the retrieving algorithm has been reported by Butz et al. (2009), Jung et al. (2016) and Connor et al. (2016). The OCO-2 data (V9r) including XCO2, CO2 profile and the a priori profile was used in this study."

2. Page 2, L41: all-weather?

Response: We corrected the sentences in line 41, Page 2 as "..., which can provide global coverage of the column-averaged dry-air mole fraction of CO2 (XCO2)"

3. P3, L92: Which one? AIMMS-20?

Response: Yes, we corrected the words "AIMMS" to "AIMMS-20AG" and we added following sentences after that: "The geolocation information including latitude, longitude, ambient pressure and height of the aircraft we used is obtained by AIMMS-20AG. In order to estimate the humidity of the environment accurately, we calculate the relative humidity by temperature and dew point. The static temperature was measured by the Total Temperature Sensor (Model 102 Type Non-De-iced, Rosemount Aerospace Inc) and the dew point temperature was measured by Dew Point Hygrometer (Model 137 Vigilant™, EdgeTech)."

4. P3, L94: Why did you use a CVI inlet? Where there other measurement (aerosol) systems onboard? Please be also more specific w.r.t. to the airborne set-up. Did you need to use an external pump to achieve the large gas flow? How long was the inlet (from the tip to the cell)? I am not aware of a publication which reports the airborne deployment of this kind of analyzer, so I suggest to include a schematic which shows

the set-up and the periphery to control cell/inlet pressure, temperature and volume or mass flows.

Response: The aircraft is designed for weather modification by China Meteorological Administration (CMA), and with lucky, we are allowed to take our greenhouse gas analyzer aboard to carry on the measurement of CO2 and CH4. The infrastructure of the aircraft and the gas flow system is designed and fulfilled in USA by the team of weather modification. We loaded our greenhouse gas analyzer inside of the aircraft and modified some gas flow arrangements to better fit the requirement for greenhouse profile measurement. We use CVI inlet and/or the ISO inlet which had been installed on the aircraft. The ISO inlet was used when the aircraft passed through the cloud, and the CVI inlet was used at other time. The schematic diagram was shown in the figure 1 (in the revised manuscript). As the schematic diagram shows, the external oil-less diaphragm vacuum pump (F-9A 08-03, GAST) was mounted between the CVI inlet and/or the ISO inlet, with the maximum pressure of 31.15 l/min used to keep a stable airflow. The length of the air channel from the tip of the inlet to the cell is about 0.6 meters. The development of an airborne system for greenhouse measurement using the cavity-enhanced absorption spectroscopy technique (CEAS) has been reported by O'Shea et al. (2013) and Palmer et al., (2013). More sentences are added in the revised manuscript in section 2.1, Line 99, Page 4: "The ultraportable greenhouse gas analyzer, UGGA (model 915-0011; Los Gatos Research), was connected to an aircraft-based impactor inlet system which consists of CVI (Model 1204; Brechtel Manufacturing Inc.) and ISO inlet (Model 1200; Brechtel Manufacturing Inc.) in the pressurized cabin for continuous measurement of CO2 and CH4."

5. P3, L96: SL/min? Which kind of Mass Flow controller?

Response: The air sample flow rate of CVI inlet is constant of 15 l/min (Aircraft-based Counterflow Virtual Impactor Inlet System CVI - Model 1204, Brochure).

The following sentences are added in the revised manuscript (Line 101, Pages 4) "The

CVI and/or ISO inlet was mounted on the top of the aircraft body as shown in figure 1 (in the revised manuscript), and the air flow rate of is kept constant by the automatic controller (Aircraft-based Counterflow Virtual Impactor Inlet System CVI - Model 1204, Brochure; Isokinetic Inlet System ISO Inlet - Model 1200, Brochure)."

6. P3, L97: The given values are from the manufacturer and might be valid for controlled laboratory conditions. Usually, the performance on a mobile platform is highly affected by variations of pressure, temperature and/or mechanical vibrations. I assume that this specific instrument is even more sensitive since it is not especially designed for use aboard research aircraft. Did you cross-check the theoretical precision values yourself in the laboratory, e.g. by supplying the system with sample gas of constant $CO_2$ and $CH_4$ mixing ratios? Did you check the sensitivity of your instrument to changes in pressure and temperature? Did you check the short and long-term drift of your instrument's sensitivity (i.e. over one flight and over a couple of days, respectively)? Did you check the repeatability of your measurements?

Response: Just before taking off, the Greenhouse Gas Analyzer (GGA) was calibrated against the standard gas, and the stability of instrument was checked and tested, immediately after touching down, again the same standard gas of $CO_2$ and $CH_4$. The data obtained after the calibration process are shown in figure 2 in the revised manuscript, it shows a relatively stable measurement and without drift after the flight. The precision and reparability of the instruments are also checked and test multiple times in the laboratory and the results show that it is stable and good for the measurements.

7. Comment: P3, L98: What do you mean with response time in this case? Is this the response time of the system to a change in atmospheric concentrations (due to e.g. the residence time in the inlet)? Is it the averaging time to achieve the given precision (in theory)? O is it the flush time of the cell and thus, gives the best achievable time resolution?

Response: The response time in this place means the averaging time to achieve the given precision, and the data processing was made to smooth with a 10-s running average to further remove errors. The residence time in the inlet from the tip to the analyzer of the system is around 220 seconds, and the data during this period in each level flight are removed (which means only the data 220 seconds after the flight keeps level are used) and reanalyzed, the modified results and figure are given in the revised manuscript.

The following sentences are added in revised manuscript (Line 104, Page 4) "The UGGA uses a cavity ringdown absorption technology, called off-axis integrated cavity output spectroscopy, to determine the trace gas concentration with a high precision of < 300 ppb ($CO_2$) and < 2 ppb ($CH_4$) and a 10-s response time according to the user manual and was tested and controlled in the laboratory."

8. P3, L98: Please specify: Where was the pressure controller installed? I assume in front of the instrument? How constant was the pressure during the flight?

Response: As the schematic diagram shows, the external oil-less diaphragm vacuum pump (F-9A 08-03, GAST) was mounted between the CVI inlet and/or the ISO inlet, with the maximum pressure of 31.15 l/min used to keep a stable airflow. There is also a smaller pump inside the UGGA system to exhaust air outside the analyzer to the outlet tube and the maximum flow of the pump of UGGA is about 0.3 l/min so the pressure and the air flow to the UGGA can be controlled. We add the cell pressure of the instrument cavity as the supplement figure 1, which shows that the pressure is stable in each level flight. The standard deviation of 0.029, 0.029, 0.033 and the range of the cell pressure of three flights is 51.31-51.43 torr, 51.32-51.43 torr, 51.30-51.42 torr on 7 August, 9 August and 10 August.

Explanation is added in the revised manuscript (Line 114, Page 4): "The instrument automatically records and saves the temperature and pressure in the cavity during measurements. The standard deviation of the cell pressure during three flights is 0.029,

0.029, 0.033 on 7, 9 and 10 August and the range of the cell pressure on each flight is below 0.12 torr."

9. P3, L99: Which temperature? The cell temperature? A range of more than 6 degrees seems very huge to me and should impact the sensitivity of the instrument. Did you check this in the lab (see also above)?

Response: Yes, it is the cell temperature, and we added supplement figure 2 shown the cell temperature with height. Considering the large variation of the cell temperature which may reduce the precision of the instrument, only data when cell temperature is within 3-sigma is used in analyses, with the range of cell temperature, respectively for the three flights, between 28.85-29.69°C, 28.26-31.37°C, 29.09-31.43°C, the standard deviation of 0.46 (7 August), 1.55 (9 August), 1.18 (10 August).

We corrected the sentences in our manuscript, Line 113, Page 4 to: "The sample cavity temperature also remained stable and constant by the temperature controller of the instrument." And we added the sentence in our manuscripts, Line 116, Page 4: "For the cell temperature, the standard deviation is 0.46, 1.55 and 1.18 on each day and the range of it is below 3.11°C."

10. P3, L99: Please provide more details on the Standards. How many standards did you use? At which concentrations? How did you calibrate the system? Did you (or some of the other institutions cross-calibrate the standards in a way that they are traceable to the typically used NIST standards? How stable was the system? How reproducible were the standard measurements before and after the flight?

Response: We added some details on the standards, and explanation is added in the revised manuscript, Line 121, Page 4: "The standard gas we used is based on dry and clean air with greenhouse gases with known concentration value, filled in a 29.5L aluminum alloy cylinder with silanization and other special treatment on the inner wall, the gas is traceable to the world meteorological organization global atmospheric observation network (WMO-GAW) level 1 standard gas (Yao et al., 2013). The concentration

of the CO2 is 400.13 ppm and CH4 is 1.867 ppm. Just before taking off, the instrument was warming up for more than 45 minutes, and then connected to the standard gas for calibration, and keep the measurement of the standard gas 5 minutes more. After landing, the standard gas is connected immediately to check the stability of the instrument by measuring the standard gas. The result is demonstrated in figure 2, and added in the revised manuscript. It can be seen that, before takeoff and after landing, the concentration is stable around the value of standard gas concentration, and there is almost no drift after the flight."

11. P4, L107, Figure 1: What do you want to show with this figure? I suggest to zoom in and to include at least the flight patterns of all 3 flights conducted in August 2017. You might also show a series of three figures with all three flight paths plotted over a weather map.

Response: We revised figure 1 (figure 3 in the revised manuscript) which shows the large area including the experiment site and the airport by google map and zooms in with flight path shown on the map. Because after taking off, the aircraft climbed up directly to the maximum height, only the paths in the decline phase are plotted to better display level flight trajectory, and only measurement during level flight are used for analysis. The flight paths for the three days is similar, so only the trajectories on 7 August is given in the article, and the flight path on 9 and 10 August aircraft are given in the supplement, we added in supplement figure 3.

We corrected the sentences in our manuscript, Line 147, Page 5 to: "Aircraft measurement were carried out from August 7 to 10 over Jiansanjiang (47.11°N, 132.66°E, 61 m above sea level), which is located in Heilongjiang province, Northeast China. Figure 2 shows the geolocation of the Jiansanjiang aircraft and the fight path."

12. Section 3: This section is especially weak. I suggest to include information about the flight strategy, as well.

Response: Thanks for the advice, we rewrite section 3, and added more sentences

about the flight strategy, and explanations are added, please see response to next comment.

13. So e.g., why did you fly in in the morning hours (during which the boundary layer develops)? Did you try to match the time of an OCO-2 overflight? Or was this due to logistical (ATC) reasons? Why did you choose to fly over a horizontal distance of 150 km (the swath is a couple of km's only)? Did you adjust the flight track to measure along-track of OCO-2? Or is this Did you always follow the same flight strategy on the three days? What was the descending rate and the corresponding pressure variation during spiraling down?

Response: The flight strategy was added and explained, and we added table 1 listed the details of the three flights. Since we cannot decide when to fly since the ATC restriction to avoid the civil aviation, so that we did not adjust the flight track to measure along-track OCO-2.

Explanation as following is added in the revised manuscript, Line 152, Page 5: "The aircraft is designed for weather modification by China Meteorological Administration (CMA), so the infrastructure of the aircraft and the gas flow system are also designed and completed in USA by the team of weather modification agency and an US company. CMA is in charge of the flight route, and there is a chance (several times later are planning) that it can carry our greenhouse gas analyzer to measure the profiles of $CO_2$ and $CH_4$. We loaded our greenhouse gas analyzer on the aircraft and modified some gas flow arrangements to better fit the requirement for greenhouse profile measurement. Due to the logistical problem and the ATC restriction, we must fly in the morning from around 7:30 to 11:00 (local time) of these days to avoid obstructing civil aviation. The details of the three flights are listed in table 1."

14. P4, L110, Figure 2: From this figure it looks like that you did ~7 constant flight legs, is that correct? I don't think you need this figure if you provide a horizontal map of the flight patterns as suggested for figure 1, which gives an idea about the flight

dimensions in Lat/Lon direction. Instead, I suggest to include a simple time-series of in-situ measured CO2, CH4, and flight altitude for this particular flight. According to figure 3, the flight trajectory in 7 August looks like that there are about 7 level flights in this flight, and the level flight is about every 300-700 m during the flight as the figure shows. From figure 3 we want to show the horizontal coverage of these flight and the flight trajectory, since the 3-D figure may not necessary in all three days which looks identical, and we added the flight trajectory of the other two days (9 and 10 August) shown in the supplement figure 3. The horizontal map overlaid with the flight trajectory in three days (7, 9, 10 August) is shown in figure 4 (in the revised manuscript), which we have revised the original one, and we hope these figures can look better which gives the idea of the dimensions of the flight pattern with relative information of the surface. And the variation of mole fraction of CO2, CH4 with flight altitude in the flights are shown in figure 5 (in the revised manuscript).

Response: We corrected the sentences and more explanations as following are added in the revised manuscript to make it clear. Line 160, Page 6: "The flight trajectory on 7 August is shown in figure 4. The aircraft climbed up quickly and directly to the maximum height to about 7.5 km 30 min after taking off, and then descending down step by step at about every 300 m. Since the 3-D figure in these three days looks identical, the flight trajectory of the other two days (9 and 10 August) is not shown in figure 4."

And also we put the information of airport Jiansanjiang and flight trajectory in the newly plotted figure 3.

15. Section 4: Please keep in mind that there are several ways how variable water vapor levels influence the CO2/CH4 measurements: 1) The dilution effect, 2) variation in the line broadening of the carbon dioxide and methane lines due to varying water vapor concentration, and 3) nonlinearity of the reported water vapor concentration due to self-broadening of the water vapor line. Here you discuss the dilution effect which certainly is the most important one. However, the water concentration measurement

must be highly accurate to allow for a meaningful accuracy in the derived dry gas concentrations. Therefore, I'd like to see an in-depth error analysis for the approach used herein. Moreover, I suggest to include the time-series of measured relative humidity and derived water vapor (including error bars!), at least in the supplement. Response: The measurement is making under real humidity conditions, so the water vapor had to be corrected to drive the CO2 and CH4 concentrations under dry conditions. We find that the measured relative humidity and temperature by AIMMS-20AG may have some uncertainty, so we used the static temperature measured by the Total Temperature Sensor (Model 102 Type Non-De-iced) and the dew point temperature measured by Dew Point Hygrometer (Model 137 Vigilant[TM], EdgeTech). To estimate the ambient humidity, we calculated the relative humidity by the dew point and temperature, and then doing water correction of CO2 and CH4 mixing ratio. We have no corresponding facilities to measure the measure the broadening effect of water vapor on the spectral line, we suppose the instrument factory have considered this effect during factory calibration experimental, so the water vapor correction except for dilution was not considered here. The RH profile we added in supplement figure 4 (a) with 1-$\sigma$ uncertainty bar, and the data is accurately time- matched to the CO2 and CH4 profiles. We also added the ambient temperature and pressure during the flight in supplement figure 4 (b) and (c), respectively. Time-series of measured pressure, dew point and temperature are given in the supplement figure 5. The data process of the meteorology data are the same as that of CO2 and CH4. The 1-$\sigma$ of the data in the level flight is taken as the uncertainty, their variation with height are also shown in the figure.

Corrections and explanation are added in the revised manuscript, Line 177, Page 6: "Where $L_v$ = 2.500$\times$106 J Kg-1, $M_w$ is the molecular weight of water equals to 18.016, R = 8.3145 J K-1mol-1, and $e_s$ (in hPa) at temperature T (in K). Pressure p (hPa) of the ambient atmosphere are measured by the aircraft meteorology system, AIMMS-20AG, and the temperature T (K) was measured by Total Temperature Sensor (Model 102 Type Non-De-iced). The relative humidity RH (%) was calculated by the dew point and

temperature. The dew point data is obtained by Dew Point Hygrometer (Model 137 Vigilant™, EdgeTech)."

16. P5, L 132: All data are recorded at 1s and then smoothed to remove errors because of the response time? As mentioned above, please be clear in the use of your wording w.r.t. response time. The residence time usually can be corrected for if e.g. volume flow and inlet pressure are known.

Response: Yes, the measurement are made at 1s frequency and then 10-s average are done to smooth and remove potential errors concerning the response time. Concerning the residence time of air flowing the pipe and cell. We have modified the data processing method to take the effect of gas residence time in the pipeline into account. Therefore, we removed the data 220s from the start during the level flight in average, because this data was acquired during the descent of the aircraft, which may cause uncertainty of the measurement.

Corrections as following are added in the revised manuscript, Line 191, Page 7: "The time points at the beginning and end of level flight are determined according to the altitude and its variation of the aircraft. Considering the residual time of the GHG measurement system, the data obtained 220 s from the start of the level flight is considered to be observed when the aircraft is descending rather than in level, which may cause uncertainty of the measurement. Therefore, the data were reserved after the level flight starting for 220 s. If the duration time of certain level flight lasted less than 220 s, the data observed during that level flight were also discarded."

17. Section 5, Figure 3: This figure shows that you did much more constant flight legs than it seems from figure 2. Do these dots represent 10s values? What is real variability and what is instrument precision? Variation in CO2 on each leg is large (maybe also because of the large horizontal distance), the vertical variability on Aug 10 seems larger than the horizontal variability on that day. Is this a real atmospheric feature and do you have any explanation for this? Also, the boundary layer variability

on 9 Aug seems much larger than on other days. Is this a horizontal gradient? Please include a more indepth discussion on these profiles. I'd like to also see the standard deviation or even better, median instead of average values including some percentiles. What was the boundary layer height? Please include also a vertical profile of met. variables, at least in the supplement.

Response: Please see the response to the previous comment about the data processing. We corrected the figure 3 as the reason mention in response to the previous comment, (figure 5 in the revised manuscript), the lines represent the average value of each level flight and the dots represent the data obtained after the 10s average, water correction and the residual time correction. The 1-$\sigma$ bars are given in the figure 6 in the revised manuscript. Since only the data during the level flight are analyzed, the data during landing time was discarded, which is about from 1 km to the surface, and it is difficult for us to correctly estimate the boundary layer height based on the observation data obtained by the aircraft. The meteorology data are given in the supplement. timeseries of temperature and dew point is shown in supplement figure 5, and the profile of RH, temperature and pressure is shown in supplement figure 4 (a), figure 4 (b), figure 4 (c), and the meteorology data are accurately time-matched with that of $CO_2$ and $CH_4$ data.

18. P5, L 142: "was attributed to different weather conditions": Please provide more details on this hypothesis.

Response: The weather condition during the three flights are sunny, overcast and overcast on 7, 9 and 10 August respectively, as the sentences in P5, L143 indicated, so we assume that the relatively larger gradient of the $CO_2$ profile from around 0.6 to 2 km on 7 August might be caused by the weaker $CO_2$ uptake from the vegetation on the surface.

We corrected the sentence as "… was probably attributed to different weather conditions…"

19. P6, L161: Please give a short introduction about the Model, the used a-priori information and the simulations – and the difference. Which data are assimilated? The OCO-2 data? Doesn't look like. The aircraft data?

Response: Tan-Tracker (v1) is a 4D-Var assimilation model and OCO-2 XCO2 retrievals (v9r) are used to optimize surface terrestrial ecosystem flux and ocean flux, with prior Fossil Fuel emission and prior Fire emission remain unchanged. Carbon Tracker posterior flux (v2017, Peters et al., 2007, https://www.esrl.noaa.gov/gmd/ccgg/carbontracker/) was used as prior terrestrial ecosystem CO2 flux and scaled ocean flux (Takahashi et al., 2009, scaled to 2016 with Marine Boundary Layer CO2 concentration www.esrl.noaa.gov/gmd/ccgg/GHGreference/) was used as prior ocean flux in Tan-Tracker (v1). Prior Fossil Fuel emission including fossil fuel emission of Open-source Data Inventory of Anthropogenic CO2 (ODIAC) (Oda and Maksyutov, 2011, http://www.odiac.org/index.html), ship emission (Endresen et al., 2007) and aviation emission of Aviation Emissions Inventory Code (AEIC) (scaled to 2016, Olsen et al., 2013). Prior Fire emission including biomass burning emission of Global Fire Emissions Database v4 (GFED4) (Randerson et al., 2018, http://www. globalfiredata.org/index.html) and biofuel emission (Yevich and Logan, 2003). The above described prior fluxes used to drive GEOS-Chem for the CO2 simulation were integrated and provided by the Harvard–NASA Emissions Component (HEMCO) model (Keller et al., 2014). Model a priori and model simulation used in our manuscript are GEOS-Chem simulation forced by prior flux and Tan-Tracker (v1) results separately. Note that prior terrestrial ecosystem flux and ocean flux are different from those used in observing system simulation experiments (OSSEs) of Han and Tian (2019). But their description of OCO-2 data assimilation experiment is still in writing, so we only cite Han and Tian (2019).

We add follow sentences in the newly added section 2.2, Line 136, Page 5, as mentioned before: "The NLS-4Dvar assimilation model Tan-Tracker (v1) and OCO-2 XCO2 (v9r) retrievals are used to optimize surface terrestrial ecosystem CO2 flux and ocean

[Figure]

CO2 flux, while prior Fossil Fuel emission and prior Fire emission remain unchanged (details of model setting and prior flux information can be found in Han and Tian, 2019)."

20. P6, L 163: "The variation of CO2 . . ..". I don't understand this sentence. Do you talk about the aircraft measurements? What about uncertainty bars for the aircraft data?

Response: The variation here means the structure of CO2 vertical profile which can be divided into three parts according to different characteristic of variation. We corrected figure 4 (figure 6 in the revised manuscript) and added 1-ïĄş bars. The sentence are corrected to make it more clear.

21. P6, L165: Reproducing CO2 uptake from vegetation by a model is highly challenging, but I do not see any information from the model (neither a-priori or simulated). Is this what you mean with "Below 2km, CO2 is assumed to be vertically mixed. . ."?

Response: Because the model keep the same mole fraction of CO2 profile below 2 km, it cannot provide any information of the source and sink on the ground. The profile of the aircraft showed low concentration near ground and increased with the height, but currently the model did not reflect this feature.

22. Comment: P6, L166: OCO-2 data were averaged over what area and what time? Please provide a graphical explanation which OCO2-Data you used. How did you get the vertical information?

Response: Because no data were obtained from OCO-2 over Jiansanjiang during the flight on 7, 9 and 10 August, we used the satellite data on 5 August which is closest in time to the experiment. The $1° \times 1°$ average of the data were used for comparison. The height information of the satellite profile is available on the satellite products. Correction and explanations as following are added in in the revised manuscript, Line 227, Page 8: "Because no data were obtained from OCO-2 (v9r) . . .."

23. P6, L175: ". . .with large differences in values". So do you have any explanation? Apart from the quality of the in-situ data, one reason might be the comparison of measurements on different days. To get an impression about the day-to-day variability in that region, you might have a look at a longer timeseries of OCO-2 data. You also might have a look at the weather conditions (low or high pressure systems, frontal crossings) and how these may have influenced the day-to-day variability.

Response: Explanations as following are added in the revised manuscript, Line 244, Page 8. "GHGs profiles have been rarely observed before near the experiment site, or over Northeast of China as far as we know. The model simulations are based on data of regional emission inventory. The accuracy of simulated profiles and concentration near surface over the experiment site still remains unknown. So continuous and regular observation of the GHGs profiles are necessary to better understand the regional emission amounts and the variation of the GHGs."

24. P7, L194: Did you use all these observations (aircore, balloon, aircraft) for your specific case or is this a general description? Is this Tan-Tracker? Please be more specific.

Response: The sentence "...high-altitude balloons, AirCore, Observations of the Middle Stratosphere balloon, and aircraft" refers to the data source used by TCCON's a prior profiles (Toon and Wunch, 2014). Due to altitude limit of flight, TCCON a prior profiles were used to extrapolate the profiles above the tropopause. As the other referee's comment mentioned, the use of any other profile will create additional biases when comparing to OCO-2 data, so we added another methods for extrapolation by using the a priori profile of OCO-2 as the supplement to the profile in the height where the measurement are not available.

Explanations as following are added in revised manuscript, Line 262, Page 9: "We used two extrapolation methods to extend the profile of the aircraft measurements and then estimates the $XCO_2$ value of the in-situ measurement respectively. 1) The unknown part of the aircraft profile was directly from the OCO-2 a prior profile. 2) A well-mixed and constant mixing ratio of $CO_2$ is assumed from the surface to the lower

limit of flight, and from the upper limit of flight to the tropopause. The CO2 concentrations above the tropopause were calculated with an empirical model (Toon and Wunch, 2014) which considers tropopause height as well as realistic latitude and time dependencies through curve fitting of data from high-altitude balloons, AirCore, Observations of the Middle Stratosphere balloon, and aircraft. In general, the mole fraction of CO2 decreased exponentially with height from the tropopause to upper stratosphere, and the tropopause height was obtained from NCEP reanalysis data with a 2.5° × 2.5° resolution, which was linearly interpolated to the geographic coordinates of Jiansanjiang. Figure 7 shows the extrapolated CO2 profiles using method (2)."

25. P7, L197: This information come much too late (should be at the beginning of section 5.2).

Response: We moved this sentence to section 5.2, Page 8, Line 258.

26. P7, L199: You compare XCO2 values with the in-situ measured data. The variability of the latter is nearly 40 ppm, which is not at all captured by the OCO-2 average profile. In my opinion, you can't compare column values and derive a bias (especially not with the accuracy given).

Response: Yes, you are right, the flight measurement was just obtained in limited altitude range, while the XCO2 is given for the whole atmosphere, therefore, they are not compared in the same level. But considering the low variation of CO2 with time in the high altitude of atmosphere, it is, at certain degree, reasonable to compare the XCO2 after the extension of the profile. To compare the uncertainty induced by the extension of the profile, results from two different extrapolation methods are used (table 4 in the revised manuscript). To assure the stable of the instrument, calibration and test against the standard gas is done just before the aircraft takeoff and checked again immediately after landing. figure 1 in the revised manuscript shows that the instrument is stable and accurate, almost no drift. After considering the residence time of the airflow in the pipeline, and removing the data in 10-second average reprocess,

the data shows much less variabilities.

27. Table 2: Please provide details on the uncertainty analysis for the aircraft errors: accuracy (traceability to WMO scale) and precision.

Response: As mentioned in previous response, calibration and test against the standard gas is done just before the aircraft takeoff and checked again immediately after landing, and the standard gas we used can be traced back to WMO scale. The average of the difference between the standard gas and the measurement of the instrument of each day was considered as the accuracy of the aircraft data. As for precision, the instrument was not continuously calibrated against the standard gas during the flight, 1-$\sigma$ deviation of the measurements during level flight of each day is considered as the precision.

And we added sentences in the revised manuscript in Page 201, Line 7: "The accuracy of $CO_2$ and $CH_4$ is below 0.66 ppm and 0.002 ppm, 0.16% and 0.10% of the $CO_2$ and $CH_4$ concentration in standard gas, respectively. For precision, the 1-$\sigma$ value is below 0.71 ppm and 0.0062 ppm for $CO_2$ and $CH_4$, respectively" The data of Table 2 (Table 4 in the revised manuscript) are corrected.

We added the following references to our manuscript: Crisp, D., Miller, C., and DeCola, P.: NASA Orbiting Carbon Observatory; measuring the column averaged carbon dioxide mole fraction from space, J. Appl. Remote Sens., 2, 023508, doi:10.1117/1.2898457, 2008. Endresen, Ø., Sørgård, E., Behrens, H. L., Brett, P. O. and Isaksen, I. S. A.: A historical reconstruction of ships' fuel consumption and emissions, J. Geophys. Res. Atmos., 112(12), 1–17, doi:10.1029/2006JD007630, 2007. Han, R. and Tian, X.: A dual-pass carbon cycle data assimilation system to estimate surface $CO_2$ fluxes and 3D atmospheric $CO_2$ concentrations from spaceborne measurements of atmospheric $CO_2$, Geosci. Model Dev. Discuss., doi:10.5194/gmd-2019-54, in review, 2019. Jung, Y., Kim, J., Kim, W. Boesch, H., Lee, H., Cho, C., and TaeYoung, G.: Impact of Aerosol Property on the Accuracy of

a CO2 Retrieval Algorithm from Satellite Remote Sensing, Remote Sens., 8, 322, doi:10.3390/rs8040322, 2016. Keller, C. A., Long, M. S., Yantosca, R. M., Da Silva, A. M., Pawson, S. and Jacob, D. J.: HEMCO v1.0: A versatile, ESMF-compliant component for calculating emissions in atmospheric models, Geosci. Model Dev., 7(4), 1409–1417, doi:10.5194/gmd-7-1409-2014, 2014. Oda, T. and Maksyutov, S.: A very high-resolution (1km×1 km) global fossil fuel CO2 emission inventory derived using a point source database and satellite observations of nighttime lights, Atmos. Chem. Phys., 11(2), 543–556, doi:10.5194/acp-11-543-2011, 2011. O'Dell, C. W., Connor, B., Bösch, H., O'Brien, D., Frankenberg, C., Castano, R., Christi, M., Eldering, D., Fisher, B., Gunson, M., McDuffie, J., Miller, C. E., Natraj, V., Oyafuso, F., Polonsky, I., Smyth, M., Taylor, T., Toon, G. C., Wennberg, P. O., and Wunch, D.: The ACOS CO2 retrieval algorithm – Part 1: Description and validation against synthetic observations, Atmos. Meas. Tech., 5, 99–121, doi:10.5194/amt-5-99-2012, 2012. O'Shea, S. J., Bauguitte, S. J.-B., Gallagher, M. W., Lowry, D., and Percival, C. J.: Development of a cavity-enhanced absorption spectrometer for airborne measurements of CH4 and CO2, Atmos. Meas. Tech., 6, 1095–1109, doi:10.5194/amt-6-1095-2013, 2013. Palmer, P. I., Parrington, M., Lee, J. D., Lewis, A. C., Rickard, A. R., Bernath, P. F., Duck, T. J., Waugh, D. L., Tarasick, D.W., Andrews, S., Aruffo, E., Bailey, L. J., Barrett, E., Bauguitte, S. J.B., Curry, K. R., Di Carlo, P., Chisholm, L., Dan, L., Forster, G., Franklin, J. E., Gibson, M. D., Griffin, D., Helmig, D., Hopkins, J. R., Hopper, J. T., Jenkin, M. E., Kindred, D., Kliever, J., Le Breton, M., Matthiesen, S., Maurice, M., Moller, S., Moore, D. P., Oram, D. E., O'Shea, S. J., Christopher Owen, R., Pagniello, C. M. L. S., Pawson, S., Percival, C. J., Pierce, J. R., Punjabi, S., Purvis, R. M., Remedios, J. J., Rotermund, K. M., Sakamoto, K. M., da Silva, A. M., Strawbridge, K. B., Strong, K., Taylor, J., Trigwell, R., Tereszchuk, K. A., Walker, K. A., Weaver, D., Whaley, C., and Young, J. C.: Quantifying the impact ofBOReal forest fires on Tropospheric oxidants over the Atlantic using Aircraft and Satellites (BORTAS) experiment: design, execution and science overview, Atmos. Chem. Phys. Discuss., 13, 4127–4181, doi:10.5194/acpd-13-4127-2013, 2013. Peters, W.,

[Figure]

Jacobson, A. R., Sweeney, C., Andrews, A. E., Conway, T. J., Masarie, K., Miller, J. B., Bruhwiler, L. M. P., Pétron, G., Hirsch, A. I., Worthy, D. E. J., van der Werf, G. R., Randerson, J. T., Wennberg, P. O., Krol, M. C. and Tans, P. P.: An atmospheric perspective on North American carbon dioxide exchange: CarbonTracker., Proc. Natl. Acad. Sci. U. S. A., 104(48), 18925–18930, doi:10.1073/pnas.0708986104, 2007. Randerson, J. T., Werf, G. R. van der, Giglio, L., Collatz, G. J. and Kasibhatla, P. S.: Global Fire Emissions Database, Version 4, (GFEDv4).ORNL DAAC, Oak Ridge, Tennessee, USA. doi:10.3334/ORNLDAAC/1293., 2018. Takahashi, T., Sutherland, S. C., Wanninkhof, R., Sweeney, C., Feely, R. A., Chipman, D. W., Hales, B., Friederich, G., Chavez, F., Sabine, C., Watson, A., Bakker, D. C. E., Schuster, U., Metzl, N., Yoshikawa-Inoue, H., Ishii, M., Midorikawa, T., Nojiri, Y., Körtzinger, A., Steinhoff, T., Hoppema, M., Olafsson, J., Arnarson, T. S., Tilbrook, B., Johannessen, T., Olsen, A., Bellerby, R., Wong, C. S., Delille, B., Bates, N. R. and de Baar, H. J. W.: Climatological mean and decadal change in surface ocean pCO2, and net sea-air CO2 flux over the global oceans, Deep. Res. Part II Top. Stud. Oceanogr., 56(8–10), 554–577, doi:10.1016/j.dsr2.2008.12.009, 2009. Yevich, Rosemarie, and Jennifer A. Logan.: An assessment of biofuel use and burning of agricultural waste in the developing world, Global biogeochemical cycles, 17(4), doi:10.1029/2002GB001952, 2003.

Please also note the supplement to this comment:
https://www.atmos-meas-tech-discuss.net/amt-2019-363/amt-2019-363-AC1-supplement.pdf

[Figure]

**Supplement:**

**The following figures are in the revised manuscript:**

[Figure]

**Figure 1. (a) The outside view of the Beechcraft King Air 350ER instrumentation. (b) The schematic diagram of the greenhouse gases sample airflow.**

[Figure]

**Figure 2. The concentration of CO2 (a) and CH4 (b) before the flight, and the concentration of of CO2 (a) and CH4 (b) after the flight obtained during the calibration, with the value of standard deviation and average of each calibration.**

[Figure]

**Figure 3. Observation area for aircraft-based measurement of CO2 and CH4 over Jiansanjiang, Northeast China, and the flight paths on 7, 9, 10 August.**

[Figure]

**Figure 4. Trajectory on the 7 August, 2018 in Jiansanjiang. The color scale shows the progression of time in local time, where blue represents the start time of the data profile, and red represents the end time.**

[Figure]

**Figure 5. Vertical profiles of (a) CO2 and (b) CH4 observed on August 7 (blue), 9 (red), and 10 (yellow), 2018, over Jiansanjiang measured in situ with aircraft. The aircraft-based in situ measurement data are indicated with dots, and averaged data for each flat flight stage are shown as lines.**

[Figure]

**Figure 6. Comparison of aircraft measurements (in situ measurement data are shown by the yellow line) with 1 standard deviation (yellow bars) collected on August (a) 7, (b) 9, and (c) 10, Tan-Tracker (v1) data (blue line) and the a priori profile of it (red line) at the location of Jiansanjiang linearly interpolated to the observation times on August (a) 7, (b) 9, and (c) 10, and the OCO-2 averaged profile (gray line) for the aircraft flight area from August 5 with 1 standard deviation (grey bars).**

[Figure]

**Figure 7. Extrapolated CO2 profiles observed on August 7, 9, and 10, 2018, over Jiansanjiang by method (2). Red, blue, and yellow solid lines show the aircraft-based (in situ) data collected on August 7, 9, and 10, respectively, averaged for each flat stage of the flight. Dotted lines show the extrapolated parts of the profiles, with colors corresponding to sampling dates in accordance with the solid lines. Black horizontal lines show the tropopause height. Black dashed lines show the PBL height.**

**The following figures are for supplement:**

[Figure]

**Supplement figure 1. The cell pressure of UGGA during the level flight on 7, 9, 10 August, 2018**

[Figure]

**Supplement figure 2. The cell temperature of UGGA during the level flight on 7, 9, 10 August, 2018**

[Figure]

**Supplement figure 3. Flight trajectory on (a) 9 August and (b) 10 August.**

[Figure]

**Supplement figure 4. (a) RH, (b) temperature and (c) pressure profiles during the flight on 7, 9, 10 August. 1-σ of the data in the level flight is taken as the uncertainty, shown as the uncertainty bar in the figure. The data process of the meteorology data are the same as that of CO2 and CH4.**

[Figure]

**Supplement figure 5. Time-series measured (a) pressure, (b) dew point and (c) temperature during the flight on 7, 9, 10 August.**

---

## Author Comment (AC2) · 14 Mar 2020

Response to Referee comment 2

The authors thank all reviewers for their constructive comments and suggestions, which have helped us to improve the quality of this paper both in sciences and writing. All comments are carefully considered and responded.

The manuscript by X. Sun et al. describes aircraft in-situ observations of CO2 and CH4 taken over Jiansanjiang, Northeastern China, between August 7 and 10, 2018. The authors used a turboprop aircraft which was limited to 0.6-7 km flight altitude. Therefore,

the profiles only covered the upper part of the planetary boundary layer (PBL) and only part of the free troposphere. In general, I greatly appreciate the efforts of taking aircraft in situ observations of CO2 and CH4 and I understand their usefulness and limitations well. However, I think the focus of the manuscript is not balanced. Due to the limited altitude coverage, the results would be most useful for validating the performance of Tan-Tracker, Carbontracker, CAMS or any other profile-based greenhouse gas data set. However, this is done only very briefly for Tan-Tracker and without much discussion about the obvious shortcomings of the model in the specific situation (active vegetation uptake of CO2 and CH4 emissions from rice fields) - especially near the surface. Instead, they spend most of the analysis and discussion on the comparison with the column-averaged OCO-2 XCO2 product - even though they correctly state that the largest error in this comparison comes from the unmeasured (extrapolated) part of their profiles. My suggestion would be to rewrite sections 5.2 and 5.3 and put more emphasis on the profile comparison. This should include a more detailed analysis how biases near the surface influence the column-averaged XCO2 and XCH4 values. Major issues: - concerning the profile to column comparison, the authors should also have a look at 1) J. Messerschmidt et al.: Calibration of TCCON column-averaged CO2 : the first aircraft campaign over European TCCON sites. Atmos. Chem. Phys., 11(21):10765– 10777, 2011. doi:10.5194/acp-11-10765-2011. 2) M. C. Geibel et al.: Calibration of column-averaged CH4 over European TCCON FTS sites with airborne in-situ measurements. Atmos. Chem. Phys., 12(18):8763– 8775, 2012. doi:10.5194/acp-12-8763-2012. Especially Geibel et al. discuss the effect of limited flight altitude on the column uncertainty due to extrapolation of the observed profiles in more detail than Wunch et al., 2010. - in Section 5.3, the authors should use the OCO-2 prior profile for extrapolating to the bottom and top of the atmosphere. The use of any other profile will create additional biases when comparing to OCO-2 data.

Thank you very much for the suggestions. We added analysis on the section 5.2 for profile comparison in our revised manuscript, Line 244, Page 8: Response: "GHGs profiles have been rarely observed before near the experiment site, or over Northeast

of China as far as we know. The model simulations are based on data of regional emission inventory. So the accuracy of simulated profiles and concentration near surface over the experiment site still remains unknown. The continuous and regular observation of the GHGs profiles are necessary to better understand the regional emission amounts and the variation of the GHGs."

We refer to the articles mentioned above, and revised extrapolation method of $CO_2$ profile in the revised manuscript. One more method is used and estimated, in the additional method, $CO_2$ concentration at altitude with no data is replaced by OCO-2 a priori profile directly.

We corrected the sentences in section 5.3 in the revised manuscript, Line 262, Page 9: "We used two extrapolation methods to extend the profile of the aircraft measurements and then estimates the $XCO_2$ value of the in-situ measurement respectively. 1) The unknown part of the aircraft profile was directly from the OCO-2 a prior profile. 2) A well-mixed and constant mixing ratio of $CO_2$ is assumed from the surface to the lower limit of flight, and from the upper limit of flight to the tropopause. The $CO_2$ concentrations above the tropopause were calculated with an empirical model (Toon and Wunch, 2014) which considers tropopause height as well as realistic latitude and time dependencies through curve fitting of data from high-altitude balloons, AirCore, Observations of the Middle Stratosphere balloon, and aircraft. In general, the mole fraction of $CO_2$ decreased exponentially with height from the tropopause to upper stratosphere, and the tropopause height was obtained from NCEP reanalysis data with a $2.5° \times 2.5°$ resolution, which was linearly interpolated to the geographic coordinates of Jiansanjiang. Figure 7 (in the revised manuscript) shows the extrapolated $CO_2$ profiles using method (2)."

And we added sentence in our revised manuscript, Line 277, Page 9: "For method (1), since the value of $CO_2$ mole faction of unknown part is the same as that of OCO-2 a-priori profile, as eq. (5) shows, no extra uncertainty would be introduced by extrapolation."

And the following sentences were added in our revised manuscript, Line 286, Page 10: "Because of the lack of observation data near the surface, the missing measurement data was directly replaced by the data at the lowest altitude measured by the aircraft. The error caused by this practice is shown in table 3, with an average of 0.79 ppm for XCO2.This is also the impact of the lack of near-surface observations on the overall XCO2 estimates. Therefore, observations from near the surface to about 1 km from other method, such as in-situ GHG measurement by tethered balloon and high tower, is necessary for accurate estimation of XCO2."

Minor issues: 1. p. 2, l. 41-42: it is not true that passive satellite observations of GHGs can provide all-weather, all-day global coverage.

We corrected the sentences in our revised manuscript, Line 41, Page 2: ". . . ,which can provide global coverage of the column-averaged dry-air mole fraction of CO2 (XCO2)."

2. p. 2, l. 51: the quantity X_gas as provided by TCCON as well as the satellite instruments is column-averaged dry-air mole fraction, not volume mixing ratio. Please check the definition of mole fraction vs. volume mixing ratio and replace "volume mixing ratio" throughout the text.

Thanks, We have checked the article and replaced the "mole fraction" and "volume mixing ration" to "column-averaged dry-air mole fraction" or "X_gas".

3. p. 3, l. 75: if possible, please provide references for all 3 satellites mentioned here.

The references are added in the article. The following 4 references for, respectively, TanSAT, GMI/GF5 and GAS/FY3D are added in the article.

We corrected the sentences in our manuscript, Line 75, Page 3: "Three satellites designed for CO2 measurement, TanSAT (Yang et al., 2018; Yang et al., 2020), GMI/GF-5 (Li et al., 2016), and GAS/FY-3D (Qi et al., 2020),. . .".

4. p. 4, l. 100: can these standard gases be referenced to the WMO GHG scale? And could you tell the nominal concentrations of CO2 and CH4 in these standards? Is

isotopic composition of the standards an issue for the aircraft measurements?

Yes, the standard gases can be tracede back to the WMO greenhouse GHG scale, which has been tested in some experiments. The concentration of the CO2 is 400.13 ppm and CH4 is 1.867 ppm of the standard. The standard gas we use has been measured in the laboratory for the proportion of $\delta$13C in CO2. The range of the proportion is -8.0‰ to -8.2‰ close to the natural content, so it will not cause significant isotopic effect on the measurement of CO2 by optical method and meet the requirements of standard gas (Yao et al., 2013).

We added the details of the standard gas in our manuscript, Line 121, Page 4: "The standard gas we used is. . .."

5. p. 4, l. 105: should be: "Aircraft measurements were carried out ..."

We corrected this sentences in the revised manuscript as, Line 147, Page 5: "Aircraft measurement were carried out from August 7 to 10 over Jiansanjiang (47.11°N, 132.66°E, 61 m above sea level), which is located in Heilongjiang province, Northeast China. Figure 2 shows the geolocation of the Jiansanjiang aircraft and the fight path."

6. p. 4, l. 109: are the mentioned times local or UTC? Please also provide the year for the dates!

Yes, the time mentioned here is local time, GMT+8.

We corrected the sentence in our revised manuscript, Line 152, Page 5, to "Three profiles were obtained between around 08:00 and 11:00 in local time (GMT+8) on August 7, 9, and 10, 2018."

7. p. 4, l. 118: be consistent in the use of mixing ratio vs. mole fraction.

We replaced all the words "mixing ratio" to "mole fraction" to keep consistent.

8. p. 5, Eq. 3: all numbers should have units in this equation!

Thanks very much, we added the units to all the numbers in the Eq.3: Lv = 2.500×106 J Kg-1, Mw is the molecular weight of water equals to 18.016, R = 8.3145 J K-1mol-1, and es (in hPa) at temperature T (in K). We corrected the sentences in our revised manuscript, Line 177, Page 6: "Where Lv = 2.500×106 J Kg-1, Mw is the molecular weight of water equals to 18.016, R = 8.3145 J K-1mol-1, and es (in hPa) at temperature T (in K)."

9. p. 5, l. 138-151: can you derive the planetary boundary layer height from your meteorological data, e.g by calculating the Bulk-Richardson number or some other indicator?

Sorry, the meteorological obtained aircraft is limited and we did not have the actual wind speed value of the atmosphere required for calculation of the Bulk-Richardson number.

10. p. 6, l. 160: the use of the word "accurate" here is misleading. If Tan-Tracker has been validated for accuracy, please provide a number. Or just drop "accurate". Besides, the aircraft observations show that the accuracy of Tan-Tracker here is limited.

Thank you, we remove the word "accurate" from the sentence.

11. p. 8, l. 219: why is one given in ppm and the other in percent?

Sorry for the sentence is misleading and the it is corrected as: "..., but the values of mole fraction of CO2 from Tan-Tracker and OCO-2 had negative bias estimates. The average bias between aircraft and OCO-2 is $-4.68 \pm 2.02$ ppm ($1.18 \pm 0.11\%$)."

12. p. 8, data availability: it would be nice if at least the 3 profiles were provided as a supplement to the paper. The amount of data should be rather small.

Yes, all the 3 profiles and relative meteorological data such as profiles of temeratureare, pressure and water vapor are available from corresponding author upon request (dmz@mail.iap.ac.cn) .

13. C3- Fig. 1: the black-and-white map of China is not very appealing. Also, a close-up of the target region, potentially as a terrain map or satellite picture would be illustrative.

Fig.1 (figure 2 in the revised manuscript)was replotted and added the flight path over the google map, we zoom in the figure focusing on the area near the experiment site.

14. - Figs. 3-5: an indication of PBL height would be useful on all these figures.

Because of the height limitation of the data, the PBL height cannot be calculated from it. So we used PBL height from reanalysis product ERA-Interim (https://www.ecmwf.int/en/forecasts/datasets/reanalysis-datasets/era-interim) with the spatial resolution of $0.125° \times 0.125°$, spatially and time averaged to the flight area and time. We revised the figure 5 (figure 8 in the revised manuscript).

15. - p. 15/16, Table 2: I assume that the numbers are for CO2 but it is not actually mentioned in the table captions. If so, a similar table for CH4 would be useful. I would also appreciate an estimate of the total resulting uncertainty.

Since OCO-2 only provide CO2 products, we do not provide the table of uncertainty of estimate XCH4. But we can provide the precision and accuracy of the CH4 measurement of the aircraft. The maximum and the average value of the difference between the standard gas and the measurement of the instrument of each day was given, and represent the accuracy of the aircraft data. For the precision, we calculated the one standard deviation of the data in each level flight, and the average and maximum value of 1-$\sigma$ on each day is considered as the precision of the aircraft measurement.

The numbers on table 2 are for CO2, and we added the it to the introduction of the table: " Table 2. Aircraft integration error budget of XCO2 estimation. . ."

We added the following references to our manuscript: Li Y. F., Zhang C. M., Liu D. D., Chen J., Rong P., Zhang X. Y., Wang S. P. CO2 retrieval model and analysis in short-wave infrared spectrum. Optik, 127, 4422-4425, doi:10.1016/j.ijleo.2016.01.144,
2016. Qi C., Wu C., Hu X., Xu H., Xu H., Lee L., Zhou F., Gu M., Yang T., Shao C., Yang Z. High spectral infrared atmospheric sounder (HIRAS): system overview and on-orbit performance assessment. IEEE Trans. Geosci .Remote Sens., PP, 1-18, doi:10.1109/TGRS.2019.2963085, 2020. Yao, B., Huang, J.Q., Zhou, L.X., Fang, S.X., Liu, L.X., Xia, L.J., Li, P.C., Wang, H.Y. Preparation of mixed standards for high accuracy $CO_2$/$CH_4$/CO measurements. Environ. Chem., 02:135-140, doi:10.7524/j.issn.0254-6108.2013.02. 019, 2013. Yang Z., Zhen Y., Yin Z., Lin C., Bi Y., Wu Liu., Wang Q., Wang L., Gu S., Tian L. Prelaunch Radiometric Calibration of the TanSat Atmospheric Carbon Dioxide Grating Spectrometer. IEEE Trans. Geosci. Remote Sens., 56, 4225-4233, doi:10.1109/TGRS.2018.2829224, 2018. Yang Z., Bi Y., Qian W., Liu C., Gu S., Zheng Y., Lin C., Yin Z., Tian L. Inflight Performance of the TanSat Atmospheric Carbon Dioxide Grating Spectrometer. IEEE Trans. Geosci .Remote Sens., PP, 1-13, doi:10.1109/TGRS.2020.2966113, 2020. Response to Referee comment 2

Please also note the supplement to this comment:
https://www.atmos-meas-tech-discuss.net/amt-2019-363/amt-2019-363-AC2-supplement.pdf

**Supplement:**

**The following figures are in the revised manuscript:**

[Figure]

**Figure 1. (a) The outside view of the Beechcraft King Air 350ER instrumentation. (b) The schematic diagram of the greenhouse gases sample airflow.**

[Figure]

**Figure 2. The concentration of CO2 (a) and CH4 (b) before the flight, and the concentration of of CO2 (a) and CH4 (b) after the flight obtained during the calibration, with the value of standard deviation and average of each calibration.**

[Figure]

**Figure 3. Observation area for aircraft-based measurement of CO2 and CH4 over Jiansanjiang, Northeast China, and the flight paths on 7, 9, 10 August.**

[Figure]

**Figure 4. Trajectory on the 7 August, 2018 in Jiansanjiang. The color scale shows the progression of time in local time, where blue represents the start time of the data profile, and red represents the end time.**

[Figure]

**Figure 5. Vertical profiles of (a) CO2 and (b) CH4 observed on August 7 (blue), 9 (red), and 10 (yellow), 2018, over Jiansanjiang measured in situ with aircraft. The aircraft-based in situ measurement data are indicated with dots, and averaged data for each flat flight stage are shown as lines.**

[Figure]

**Figure 6. Comparison of aircraft measurements (in situ measurement data are shown by the yellow line) with 1 standard deviation (yellow bars) collected on August (a) 7, (b) 9, and (c) 10, Tan-Tracker (v1) data (blue line) and the a priori profile of it (red line) at the location of Jiansanjiang linearly interpolated to the observation times on August (a) 7, (b) 9, and (c) 10, and the OCO-2 averaged profile (gray line) for the aircraft flight area from August 5 with 1 standard deviation (grey bars).**

[Figure]

**Figure 7. Extrapolated CO2 profiles observed on August 7, 9, and 10, 2018, over Jiansanjiang by method (2). Red, blue, and yellow solid lines show the aircraft-based (in situ) data collected on August 7, 9, and 10, respectively, averaged for each flat stage of the flight. Dotted lines show the extrapolated parts of the profiles, with colors corresponding to sampling dates in accordance with the solid lines. Black horizontal lines show the tropopause height. Black dashed lines show the PBL height.**

**The following figures are for supplement:**

[Figure]

**Supplement figure 1. The cell pressure of UGGA during the level flight on 7, 9, 10 August, 2018**

[Figure]

**Supplement figure 2. The cell temperature of UGGA during the level flight on 7, 9, 10 August, 2018**

[Figure]

**Supplement figure 3. Flight trajectory on (a) 9 August and (b) 10 August.**

[Figure]

**Supplement figure 4. (a) RH, (b) temperature and (c) pressure profiles during the flight on 7, 9, 10 August. 1-σ of the data in the level flight is taken as the uncertainty, shown as the uncertainty bar in the figure. The data process of the meteorology data are the same as that of CO2 and CH4.**

[Figure]

**Supplement figure 5. Time-series measured (a) pressure, (b) dew point and (c) temperature during the flight on 7, 9, 10 August.**

---

## Referee Report (RR1)

The paper by Xiaoyu Sun et al. describes airborne in situ measurements of two major greenhouse gases (GHGs), CO2 and CH4, in Northeast China on 3 days in August 2018. Using a twin engine turboprop aircraft, GHG mole fractions were sampled between altitudes of 0.6 km – 7 km above a predominantly agricultural area. Three flights, each consisting of spiral-down maneuvers with intermittent constant-altitude segments, were carried out in the morning hours, between 0800 and 1100 local time. The authors provide quasi-instantaneous vertical profiles from these three flights and compare to carbon cycle data assimilation system (Tan-tracker) and OCO-2 profiles. The subject of this manuscript is highly topical and targets the need for high-quality, traceable airborne observations of the main GHGs to tackle uncertainties in the global carbon cycle. I recommend publication after addressing some minor comments.

*General:*
*Sect. 2:* Have there been any in-flight measurements of the standard gases? Unfortunately it is often not enough to calibrate the instrument before and after the flight, as the ambient conditions during flight induce spectroscopic changes that can only be verified from in-flight online calibration. The instrument may well be stable right before and after the flight, which however does not imply stability under flight conditions. This is presumably also the reason why the authors omitted data associated with rapid vertical movement.

Sect. 4.1: I wonder why the authors refrained from using the co-measured H2O mole fractions. To my knowledge the UGGA 915-0011 provides water vapor measurements that are implicitly used to online correct for dilution and broadening effects.

*Specific:*
p. 2, l. 33-35 Please provide a reference for the statement "Accurate measurements of […]"

p. 5, l. 161 What does the word "reserved" refer to in "[…] data collected and reserved while […]"

p. 8, l. 219 The authors can only claim this for the 3 days of measurement and the region sampled.

p. 8, l. 228 How many stacked layers reside in the vertical coverage of the aircraft?

p. 8, l. 240 Please be more specific about "[…] but with large differences in values."

---

## Author Response (AR2)

**Response to Reviewer Comments of reviewer 1**

*The authors thank all reviewers for their constructive comments and suggestions, which have helped us to improve the quality of this paper both in sciences and writing. All comments are carefully considered and responded. The response in blue italic letters follow each comment in black.*

- in your response to my comments you write that "We refer to the articles mentioned above, and revised extrapolation method of CO2 profile in the revised manuscript." I suppose "the articles mentioned above" refers to Messerschmidt et al., 2011 and Geibel et al., 2012 that I suggested? In your revised manuscript, they are not listed in the updated reference list. So if you used them, please add them there.

*Answer: Thanks for the correction. Yes, the articles I mentioned and refer to are those you suggested. I have added the two articles in the reference list in the revised manuscript.*

- please make sure that the use of dates is consistent. Especially Tables 1 and 2 use "7 August" and "August 9/10". I found a few similar issues in the text.

*Answer: I have modified the use of dates, and unified it in the whole article as "7 August".*

**Response to Reviewer Comments of reviewer 2**

*The authors thank all reviewers for their constructive comments and suggestions, which have helped us to improve the quality of this paper both in sciences and writing. All comments are carefully considered and responded. The response in blue italic letters follow each comment in black.*

General: Sect. 2: Have there been any in-flight measurements of the standard gases? Unfortunately, it is often not enough to calibrate the instrument before and after the flight, as the ambient conditions during flight induce spectroscopic changes that can only be verified from in-flight online calibration. The instrument may well be stable right before and after the flight, which however does not imply stability under flight conditions. This is presumably also the reason why the authors omitted data associated with rapid vertical movement.

*Answer: Thanks for the comments. Yes, in-flight calibration is very crucial for high quality data, but at this moment, calibration processes is not applied during flight, because standard gas compressed in cylinder is not allowed to be aboard on the aircraft. Other than the aboard permission, the time duration of the calibration is relative longer to make for each level. As your reminding, we are thinking to do such kind of in-flight calibration in the future field measurements.*

Sect. 4.1: I wonder why the authors refrained from using the co-measured H2O mole fractions. To my knowledge the UGGA 915-0011 provides water vapor measurements that are implicitly used to online correct for dilution and broadening effects.

*Answer: Thank you for the comment. Yes, as you say, the instrument has the ability for co-measuring $H_2O$ mole fractions. As we have no standard gas for humidity calibration, and we are not sure of its precision, the co-measured H2O data is not used in this paper. But we agree with you that the using of online measurements for dilution correction is much better, and another high-precision sensor for water vapor will be integrated with*

*this instrument for calibration and high-precision measurement. Your comments will be*

*adopted in the future measurements.*

Specific: p. 2, l. 33-35 Please provide a reference for the statement "Accurate measurements of […]"

*Answer: Thanks, we added the following references to the revised manuscript:*

*Zhang, D., Tang, J., Shi, G., Nakazawa, T., Aoki, S., Sugawara, S., Wen, M., Morimoto, S., Patra, P. K., and Hayasaka, T.: Temporal and spatial variations of the atmospheric $CO_2$ concentration in China, Geophys. Res. Lett., 35, http://doi:10.1029/2007GL032531, 2008.*

*Patra, P. K., S. Maksyutov, M. Ishizawa, T. Nakazawa, T. Takahashi, and J. Ukita (2005a), Interannual and decadal changes in the sea-air $CO_2$ flux from atmospheric $CO_2$ inverse modeling, Global Biogeochem. Cycles, 19, GB4013, doi:10.1029/2004GB002257.*

*Patra, P. K., M. Ishizawa, S. Maksyutov, T. Nakazawa, and G. Inoue (2005b), Role of biomass burning and climate anomalies for land atmosphere carbon fluxes based on inverse modeling of atmospheric $CO_2$, Global Biogeochem. Cycles, 19, GB3005, doi:10.1029/2004GB002258.*

*These referenced are also cited in context, P2, L35:*

*"… models (Patra et al., 2005a, 2005b; Zhang et al., 2008)."*

p. 5, l. 161 What does the word "reserved" refer to in "[…] data collected and reserved while […]"

*Answer: Thanks for the correction, the words "and reserved" are deleted and the sentence is corrected as "…only data collected while spiralling downward were regarded as valid…." in P5, L161*

p. 8, l. 219 The authors can only claim this for the 3 days of measurement and the region sampled.

*Answer: Thanks, we corrected the sentences in our manuscript, P8, L 218 to:*

*"The results show that the vertical profile of $CO_2$ in summer increases with height in the upper troposphere, whereas that of $CH_4$ changes little with height and is relatively stable over Jiansanjiang area during experiment."*

p. 8, l. 228 How many stacked layers reside in the vertical coverage of the aircraft?

*Answer: The number of the stacked layers of the aircraft are 17, 21 and 20, respectively, in 7, 9 and 10 August.*

*We add a sentence after line 164 to address this information.*

p. 8, l. 240 Please be more specific about "[…] but with large differences in values."

*Answer: We delete the word "large" to avoid misunderstanding and added the following sentence in our manuscript, P8, L 240:*

[revised manuscript text omitted]